# Non-trivial stimuli-responsive collective behaviours emerging from microscopic dynamic complexity in supramolecular polymer systems

Martina Crippa[1], Claudio Perego [2] ✉ & Giovanni M. Pavan [1,2] ✉

Supramolecular polymers are composed of monomers that self-assemble non-covalently generating distributions of fibres in continuous exchange-and-communication with each other and the surroundings. Intriguing collective properties may emerge in such molecular-scale complex systems, following mechanisms often difficult to ascertain. Here we show how non-trivial collective behaviours may emerge in dynamical supramolecular polymer systems already at low-complexity levels. We combine minimalistic models, simulations, and advanced statistical analyses investigating how cooperative and non-cooperative supramolecular polymer systems respond to a specific stimulus: i.e., the addition of molecular sequestrators perturbing their equilibrium. Our data show how, while in a non-cooperative system all assemblies populating the system suffer uniformly the perturbation, in a cooperative system the larger/stronger assemblies survive at the expense of the smaller/weaker entities. Collective behaviours typical of larger-scale and more complex (social, economic, etc.) systems may thus emerge even in relatively simple self-assembling systems from the internal (microscopic) dynamic heterogeneity of their ensembles.

Supramolecular polymers, formed by monomers that self-assemble *via* reversible non-covalent interactions, such as, e.g., hydrogen bonding, solvophobic repulsion, $\pi - \pi$ stacking, etc.[1,2], have attracted considerable attention due to their dynamic properties and potential applications for the development of advanced materials[1,3–5]. The reversibility of the constitutive intermolecular interactions are at the basis of the non-trivial, adaptive response of such materials to external stimuli[1,6–9]. The response of supramolecular polymers to various types of stimuli, such as, e.g., temperature changes[10–12], light irradiation[13,14], dilution[15,16], etc., has been widely explored. For example, the addition of secondary chemical species, which are able to interact and co-assemble with the monomers that generate the supramolecular

polymer, can give rise to collective phenomena and systems' adaptations[15,17–22]. A notable example of such molecular perturbations is the addition of molecular chain-stoppers, which can bind onto the free extremities of supramolecular polymers, saturating their binding sites (the fibre tips), and inhibiting fibres' growth[23–26]. The addition of chain-stoppers may strongly impact the collective properties of the system[13,24–32], for instance, by reducing the average polymerisation degree, as also shown by theoretical mass-balance-based models[23,26,27,33]. Experiments also revealed a complex interplay between the self-assembling monomers and the chain-stoppers[31], which may give rise to nontrivial processes such as, e.g, dilution-induced polymerisation[8,16,28]. These phenomena are typically not easy

[1]Department of Applied Science and Technology, Politecnico di Torino, Torino, Italy. [2]Department of Innovative Technologies, University of Applied Sciences and Arts of Southern Switzerland, Polo Universitario Lugano, Campus Est, Lugano-Viganello, Switzerland. ✉e-mail: claudio.perego@supsi.ch; giovanni.pavan@polito.it

to understand because, in general, the response of supramolecular polymers to perturbations involves the adaptation of a complex network of molecular/supramolecular interactions, which cannot be easily captured using average approaches that overlook the microscopic dynamics of these systems.

Elucidating how supramolecular polymer systems respond to external stimuli or perturbations, such as the addition of chain-stopper molecules, requires answering general questions such as e.g.,: do all entities (i.e., different size oligomers/fibres) populating the supramolecular polymer systems respond in the same way to the effect of the stimulus? What factors determine how the system responds to the external perturbation? Where does a collective stimuli-responsive behaviour originate from? Answering these questions demands in-depth investigations and, in particular, a change of perspective in how stimuli-responsiveness is analyzed: instead of focusing on individual assembled objects, the response of the system must be considered as an emergent property of the entire polymer ensemble. This requires overcoming the typical estimations of average properties, moving toward a full, microscopic-level characterisation of the dynamical interaction networks proper of such complex systems[34]. Specifically, it is necessary to evaluate how such dynamical network, that involves all molecular and supramolecular entities in the system, changes in response to the stimulus. This is, in most cases, prohibitively difficult experimentally.

Molecular models and simulations are fundamental to this end[34–39]. In particular, minimalistic coarse-grained (CG) models that are developed to retain the characteristic properties of realistic supramolecular polymer systems, allow simulating space and time scales relevant to capture the equilibrium inter-assembly dynamics of model systems containing many interacting fibres/oligomers[34,38,39]. This provides insights into the structural and dynamical features of the supramolecular ensemble, with great detail on the molecular exchange dynamics of supramolecular polymer systems. Using such approaches, we recently proposed a new viewpoint to characterise this *supramolecular dynamical equilibrium* as that of complex systems[34], showing how even the simplest dynamical self-assembling system can give rise to a complex dynamical communication network, which interconnects all the entities. This demonstrates how the slightest change in the structure/features of monomers can produce substantial changes at the ensemble-level for what pertains the structure and dynamics of the system.

Here, we leverage on this complex molecular systems simulation paradigm to elucidate how supramolecular polymer systems are impacted when subject to a chemical perturbation, focusing on the notable case of chain-stopper addiction. Employing minimalistic CG models representative of two prototypical cooperative and non-cooperative supramolecular polymer systems, we explore how these two systems respond to the addition of chain-stoppers that perturb their inherent equilibrium molecular communication networks. Detailed statistical analyses of the molecular dynamics trajectories demonstrate how in these systems stimuli-responsiveness originates from the heterogeneity of their dynamic communication network. The results also demonstrate how complex collective properties, that are typically attributed to larger scale and more complex systems (for example, social systems, economic systems, etc.) may originate even in relatively simple self-assembling systems, provided that they possess an internal dynamical heterogeneity. Altogether, we come out with new concepts useful to understand how to rationally design and program emergent properties in self-assembling systems.

## Results

### Internal complexity of cooperative and non-cooperative supramolecular polymer systems

We start by considering CG, minimalistic monomer models forming to supramolecular fibres via self-assembly. We focus on the CG models of two supramolecular systems: the non-cooperative (isodesmic) supramolecular polymer model presented in Crippa et al.[34], that we name **M** model, and a cooperative variant of this model, named **M**$_{coop}$. These **M** and **M**$_{coop}$ minimalistic models feature stochastic dynamics and simple monomer-monomer interactions that, despite the approximations, allow to qualitatively represent the essential features in terms of structural and dynamical behaviours of real supramolecular polymer systems (such as, e.g., BTA, BTT, porphyrin-based polymers in good solvents)[34,38,40]. While the simplicity in the physical description provided by these models does not preserve a direct correspondence to specific chemical systems, it is useful to extend the time- and space-scales accessible by the MD simulations, and to attain general-level insights into collective dynamical behaviours of such complex self-organising systems.

The **M** monomers are planar, hexagonal molecules made of seven particles in total: a central core particle, and six shielding ones (Fig. 1a) providing directionality to the self-assembly. The **M** monomers self-assemble into 1-dimensional stacks *via* the non-bonded attractive Lennard-Jones (LJ) interactions between central beads (Fig. 1a, black), while other beads (gray) act as excluded volume. Such rigid model essentially follows an isodesmic polymerisation scheme: i.e., a self-assembling system where the monomer association constant is independent of the fibre length. The cooperative monomer variant **M**$_{coop}$ has topology and self-assembling propensity consistent with those of the **M** model, but the monomer-monomer interactions are different. The central beads (red) in **M**$_{coop}$ are attracted *via* a weaker LJ interaction than in the **M** model ($\epsilon = 20$ kJ mol$^{-1}$), while a rigid freely-rotating dipole formed by two oppositely charged beads (Fig. 1b, $q = \pm 1.4\,e$) is anchored at the center of the monomer (see "Methods" section for further details). The assembly of **M**$_{coop}$ in stacks is favoured by the alignment of these rotating dipoles along the orthogonal axes of the hexagonal monomers. This introduces an entropic penalty to dimerisation, due to the relatively low likelihood that two free monomers stack with the dipoles oriented to favour binding. This is shown in the Supplementary Information (SI), Fig. S12a–c, which report the free-energy of dimer formation in the **M** and **M**$_{coop}$ systems, evaluated via Well-Tempered metadynamics[37,40,41], and its decomposition into enthalpic and entropic terms (see "Methods" section for details). The polymerisation of longer oligomers/fibres, in which the dipoles at the ends are already aligned is thus favoured compared to that of monomers/smallest assemblies, where dipoles have random or weak alignment and their interactions are inherently unstable and more dynamic (see Fig. S12d). This confers to the **M**$_{coop}$ model the typical features of a cooperative supramolecular system, such as, e.g., the presence of a critical nucleus for fibre elongation, and a characteristic bimodal size distribution (see below).

We compare two systems of 2000 monomers (either **M** or **M**$_{coop}$), at constant volume and temperature of $T = 300$ K. The monomers, randomly distributed at the beginning, equilibrate after a transient self-assembly stage (of few μs, see Fig. S1), forming fibres of various sizes (insets of Fig. 1a,b). Once equilibrium is reached, the stacking order $\phi$ (i.e., the average coordination between the central beads of the monomers) in the **M**$_{coop}$ system matches that of the **M** system (see Fig. 1d). Figure 1c shows the distribution of sizes of the fibres that populate the systems at equilibrium. Despite the two systems reach an identical average assembly size (1c, blue line), the **M**$_{coop}$ system displays a peak of free monomers that is not present in the non-cooperative **M** system. The abundance of free oligomers (assemblies smaller than a critical nucleus, estimated of size 3–4 monomers in the **M**$_{coop}$ system) is typical of cooperative systems, where the association constant depends on the size. In Fig. 1e we report the distribution of different-size assemblies that populate the system, showing that a microscopic dynamic equilibrium is established in both systems: the sizes distributions follows Fig. 1c,

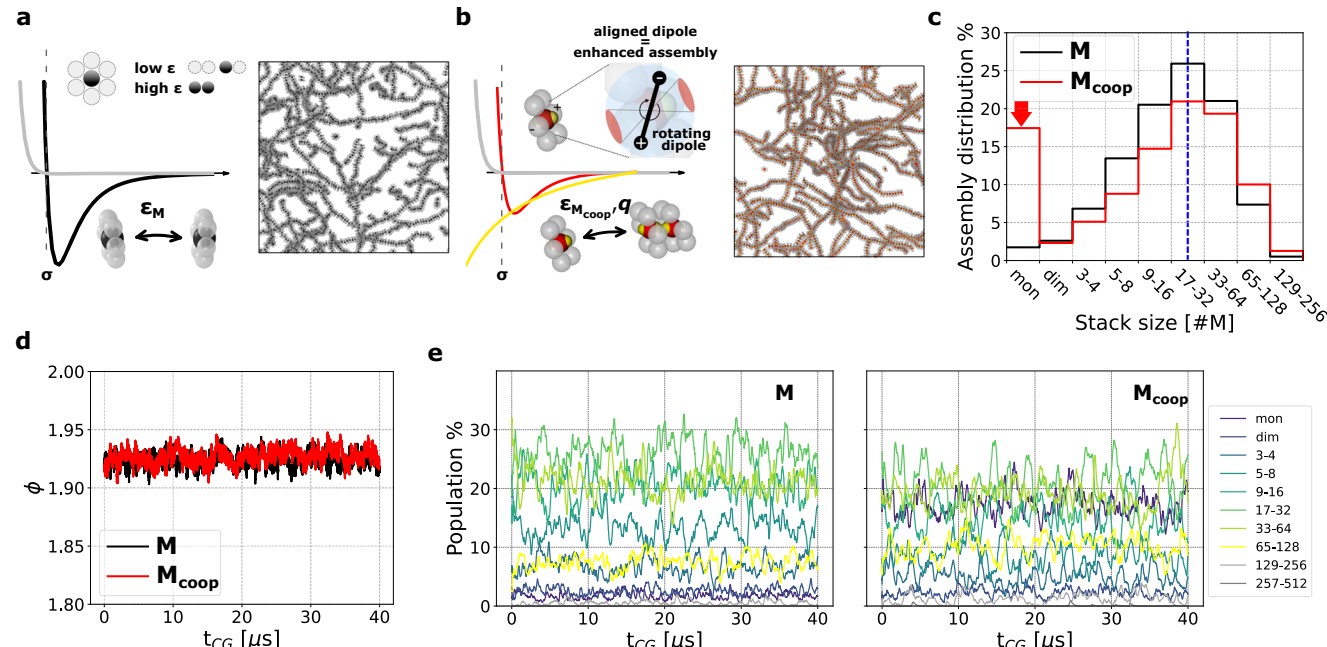

**Fig. 1 | Comparison of M and M$_{coop}$ systems. a** Left: Structure and interaction of the **M** minimalistic model: the monomers interact directionally *via* attractive interaction between the central black beads (black curve). Gray beads interact repulsively (gray curve) to screen the central beads, enforcing aggregation along the axis. Right: CG-MD simulation snapshot of a **M** model system of 2000 monomers in equilibrium (at *T* = 300 K). **b** Left: Structure and interaction of the **M$_{coop}$** minimalistic model: the monomers interact directionally *via* attractive interaction between the central red beads (red curve) and *via* dipole-dipole interactions generated by a free rotating dipole positioned at the monomer center (the Coulomb interaction between two opposite charges, yellow beads, is reported in yellow). Right: CG-MD snapshot of a **M$_{coop}$** model system of 2000 monomers in equilibrium (at *T* = 300 K). **c** Left: Assembly-size distribution (in % of the average number of assemblies), for the **M** (black) and **M$_{coop}$** (red) models, the sizes are grouped in log binary scale. The blue dashed line indicates the average equilibrium size and the red arrow highlights the cooperative monomer peak. **d** Parameterisation of **M$_{coop}$** force field: the interaction strength is set to $\epsilon$ = 20 kJ mol$^{-1}$ to match the average coordination $\phi$ of the **M** system. **e** Equilibrium distribution of assemblies of different size, along simulation time.

but a continuous molecular exchange is evident from the fluctuations of the population percentage associated to different sizes.

To characterise the molecular exchange dynamics along the equilibrium stage of the simulations we keep track of the individual trajectories of all monomers, identifying at each step their *aggregation state*, i.e., the size of the aggregate they belong to (monomer, dimer, and so on). This enables the tracking of all the transition events involving the monomers (all the changes in their aggregation state). On a global level, this allows to compute the molecular traffic, which sums all supramolecular binding/unbinding events over time[42]. The traffic increases linearly in both systems, as a result of equilibrium dynamics (see Fig. 2a), but it grows ~40% more rapidly in the **M** system compared to the **M$_{coop}$** system, showing that the former is on average more dynamic. Tracking all monomer exchange events allows us also to enter in detail into the transitions of different-size fibres, reaching a microscopic-level description of the systems' supramolecular equilibrium dynamics. We could therefore evaluate the transition matrices associated to the two systems (Fig. S2), counting all transition events that connect all possible aggregation states, i.e., the molecular exchange between all the different-size assemblies that populate the system at equilibrium. Via normalisation of the transition matrices rows, we then extract the transition probabilities among different aggregation states, i.e., the probability of a monomer being in an assembly of size *i* to go into an assembly of size *j*, within a certain time interval (in the analysis shown herein $\Delta\tau$ = 15 ns, see "Methods" section). Figure 2b gathers such transition probabilities into matrices, reporting polymerisation (upper-triangular entries), persistence (diagonal) and depolymerisation (lower-triangular) probabilities. These matrices provide important indications on the networks of communication between all the molecular entities[34], unveiling the

differences between the two systems' microscopic dynamics. While, on average the **M$_{coop}$** shows lower dynamicity than the non-cooperative **M** system, oligomers smaller than the critical size (~3–4) are highly dynamic and unstable (i.e., they tend to disassemble into monomers with high probability).

In Fig. 2c we gather the matrices data in polymerisation and depolymerisation probability curves (respectively $P_{poly}$ and $P_{depoly}$, see "Methods" section) as a function of the aggregate size. The curves showcase a qualitatively different behaviour between the two models, signaling the presence of a critical nucleation/polymerisation size in the **M$_{coop}$** system, typical of cooperative systems. The ratio $P_{poly}/P_{depoly}$ shown in Fig. 2d indicates the balance between polymerisation and depolymerisation dynamics at the equilibrium. This quantity displays a non-monothonic behaviour in the **M$_{coop}$** system, which identifies a critical size of ~3–4, whereas in the **M** system the curve decreases monotonically with size.

To further enrich this characterisation, we performed an analysis of the microscopic dynamics employing the LENS descriptor[43], which quantifies the reshuffling in the neighbourhood of each molecule at each time-interval $\Delta t$ (here 300 ps). In this kind of systems, local reshuffling is a clear fingerprint of the supramolecular communication dynamics, indicating how monomers are exchanged across the fibres structure. We report the full LENS analysis in the SI (Fig. S3), while in Fig. 2e we show the main transition mechanisms identified. The events correspond to the polymerisation/depolymerisation of a monomer (purple), the polymerisation/depolymerisation of a fibre (green), the bending/straightening of the fibre backbone (blue) and the formation/rupture of a branching (orange). The first one is the unique event involving exchange of a single monomer, while the others involve multiple molecules (e.g fragmentation and recombination). The results show that in the **M** system the fragmentation/recombination

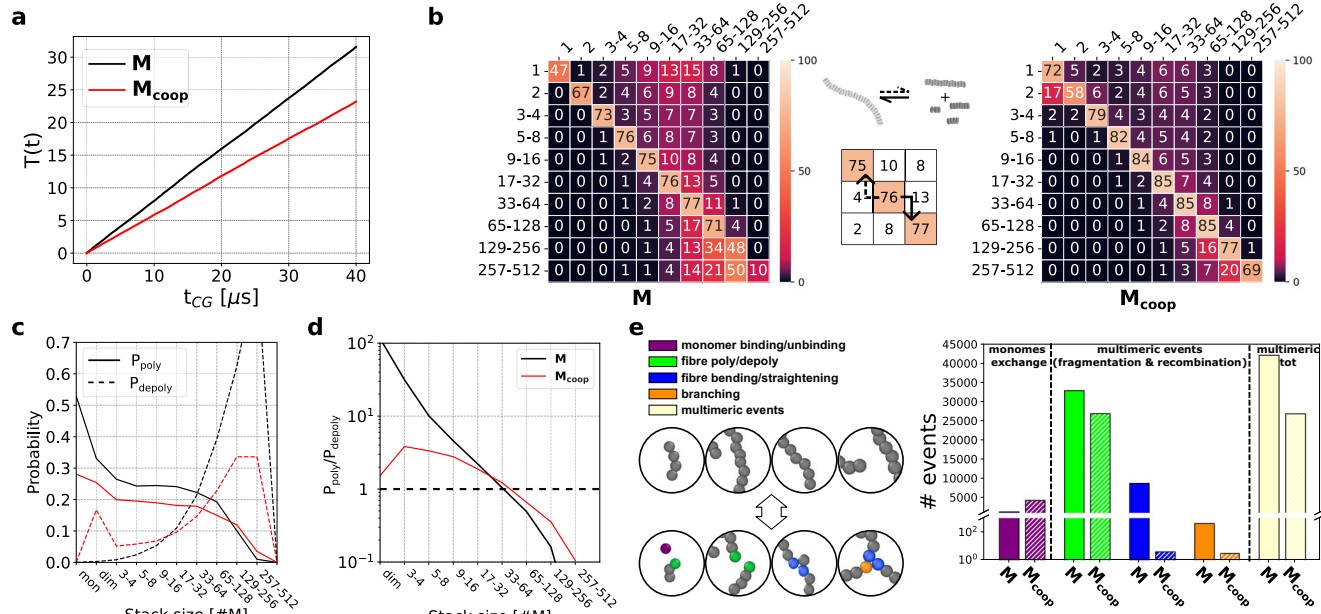

**Fig. 2 | Equilibrium dynamics of M and $M_{coop}$ models. a** Molecular traffic $T(t)$ of **M** and $M_{coop}$ systems at the equilibrium. **b** Transition probability matrices (probability in %, with $\Delta\tau = 15$ ns) of **M** and $M_{coop}$ systems. The diagram in the middle indicate the pathways defined by polymerisation (solid) and depolymerisation (dashed) of an assembly across the matrix. **c** Polymerisation and depolymerisation probability curves as a function of aggregate state (computed with $\Delta\tau = 15$ ns). **d** Polymerisation/depolymerisation ratio as function of aggregate size (computed with $\Delta\tau = 15$ ns). **e** Dynamic events detected by LENS descriptor[43] for the **M** and $M_{coop}$ systems at the equilibrium. Four examples are reported herein (see Fig. S3

for the full analysis). Each event connects the conformations shown in the upper and lower circles: binding/unbinding of a free monomer (the purple monomer binds/unbinds from an existing fibre), polymerisation/depolymerisation of an existing fibre (binding/unbinding of green tip monomers), fibre bending/straightening (the blue monomers form/break a contact out of the fibre axis) and branching (orange monomer binds/unbinds with two blue monomers out the fibre axis). (right) Number of events detected for each type, the dashed line separates multimeric and monomeric events. The last columns (cream) indicate the total number of multimeric events.

events are more prominent than in the $M_{coop}$ system, also indicating a lower persistence length of the **M** fibres compared to the $M_{coop}$ ones. On the contrary, events involving monomers are more numerous in the $M_{coop}$ system. In general, cooperativity increases the dynamic non-uniformity of the system: larger assemblies become stronger and less dynamic, while smaller assemblies become weaker and more dynamic. We also performed the same comparison between smaller **M** and $M_{coop}$ systems, containing 500 monomers at the same concentration (1/4 cell volume). Such smaller size systems report the same distinctive features for the cooperative ($M_{coop}$) and non-cooperative (**M**) systems, proving that the observed behaviours are not significantly impacted by finite-size effects (see Fig. S7).

Furthermore, to enrich the comparison between the **M** and $M_{coop}$ systems we provide further details on their diffusion and rotation dynamics at the equilibrium, and on the effect of the formation of the polymer network. The results, reported in the SI (Fig. S10), show that the dipoles have a small impact on the dynamics of free monomers, but significant differences between **M** and $M_{coop}$ dynamics emerge when the equilibrium fibre distribution is formed.

Overall, the shown comparison highlights how radically different the supramolecular dynamics of the **M** and $M_{coop}$ systems is at the equilibrium, despite they reach the same polymerisation degree. This evidence is an ideal baseline to question how supramolecular systems characterised by different cooperativity respond to external stimuli, considered that the ensemble response emerges directly from such dynamic networks of supramolecular communication.

### A microscopic view of stimuli-responsiveness
Typically, when a supramolecular polymer system in equilibrium is perturbed by an external stimulus (addition/subtraction of energy), it moves to a non-equilibrium state, and subsequently evolves toward a new equilibrium. To study stimuli-responsiveness, we here explore the

case in which chemical energy is introduced in the system in the form of an extra species that perturbs the structural and dynamical equilibrium of the supramolecular system. Specifically, it has been shown that adding chain-stoppers, molecules similar to the constitutive monomers but *capped* on one side, affects the dynamic equilibrium of a supramolecular polymer system[13,24–32]. Such chain-stoppers can bind the fibre tips, inhibiting polymerisation at those sites, and disrupting the supramolecular equilibrium underpinning the polymerisation/depolymerisation balance.

We developed a minimalistic model of a chain-stopper (**C**) species, that interacts with the **M** and $M_{coop}$ units. The **C** monomers are analogous of the **M** monomers, but they have an additional, out-of-plane shielding particle (gray) that inhibits the growth of the fibre once the **C** unit binds to a fibre tip (Fig. 3a). The open site of **C** interacts *via* a LJ potential with the cores of other **C, M** or $M_{coop}$ units (see also "Methods"). The **C** interaction strength is the same with all three counterparts (equal to the **M**-**M** interaction: see "Methods"). The chain-stopper species thus interacts equally with **M** or $M_{coop}$ fibres, while the association constant of **C** monomers to other aggregates is independent of the size of the latter (the **C**-interaction is non-cooperative). As shown in Fig. 3a, when one **C** binds a fibre tip, the polymerisation is prevented, and only depolymerisation events are possible.

We explored the response of **M** and $M_{coop}$ systems to the insertion of different amounts of **C** monomers into their equilibrium configurations. As shown in Fig. 3b, the addition of **C** reduces the stacking order $\phi$ proportionally to the amount of added **C** molecules. In all cases, the average order parameter $\phi$ relaxes to a new equilibrium on a time scale of 1 -30–40 µs after **C** insertion. The Sankey diagrams of Fig. 3c show how the communication networks that characterise **M** and $M_{coop}$ systems adapt dynamically to the perturbation (200 **C** in this case). This figure highlights how, after **C** addition, these systems evolve toward a new equilibrium state, characterised by a network of

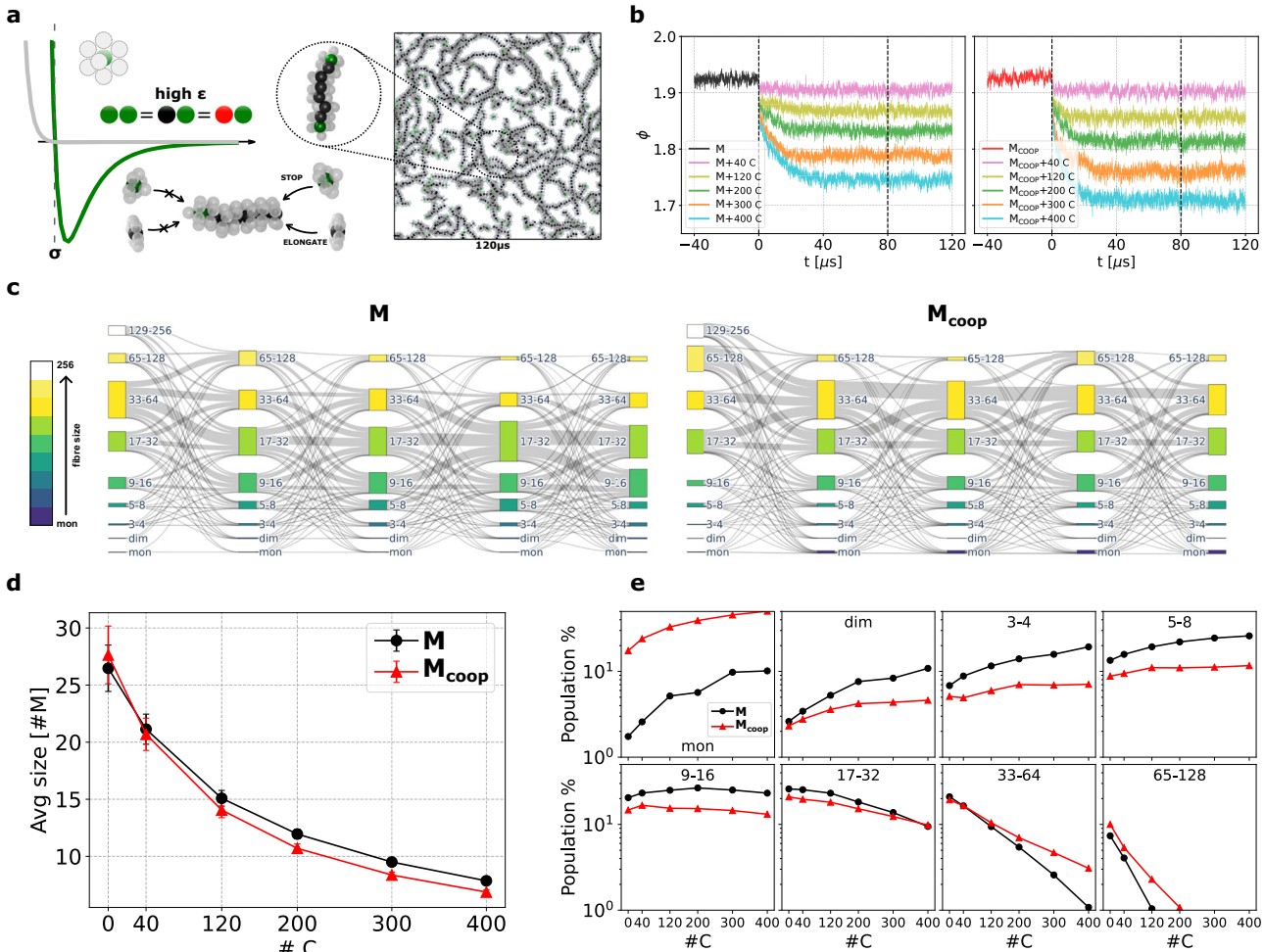

**Fig. 3 | Structural response of M and M$_{coop}$ systems to chain-stoppers insertion.**
**a** Left: Structure and interaction of the chain-stopper minimalistic model **C**: the monomers interact directionally *via* LJ attractive interaction between the central beads (of the **C**, **M**, or **M$_{coop}$** monomers). The extra shielding bead of **C** prevents further elongation of the fibres. Right: CG-MD simulation snapshot of a model system composed of 2000 **M** monomers and 120 **C**, the inset highlights the **C** binding to an **M** fibre. **b** Average coordination of **M** (left) and **M$_{coop}$** (right) upon chain-stopper insertion. The unperturbed equilibrium behaviour ($t < 0$) is affected at $t = 0$ with the insertion of different amounts of **C** (see legend). At $t = 80\,\mu s$ the

systems can be safely considered in equilibrium with the added **C**. **c** Sankey diagrams showing the transitions of monomers between the different size aggregates at $t = 0, 5, 10, 20$ and $50\,\mu s$, focusing on the equilibration part of the **M** and **M$_{coop}$** systems, after the insertion of 200 **C**. **d** Average size at the equilibrium as a function of the number of inserted **C**. The error bars show the standard deviation of the average size. **e** Equilibrium prevalence of different assembly sizes (in % of the average number of assemblies) as a function of the **C** number. For simplicity, assembly sizes are grouped logarithmically.

interactions/exchange connecting the different aggregate populations that differs from the native one.

The hindering of fibre growth imposed by chain-stoppers induces a reduction in the average polymer size at the equilibrium, in agreement with previous experience[24,26]. In our simulations we observe that this reduction is similar in the **M** and **M$_{coop}$** systems (Fig. 3d). However, more interestingly, the impact on the size-distribution, i.e., the variations in the distribution of monomers among different aggregate states (monomers, dimers, etc.)- is significantly different in the two systems. Figure 3e shows the effect of varying the **C** concentration on the average amount of different-size assemblies populating the new equilibrium states. In both systems, the concentration of assemblies smaller than the unperturbed average size (~26–27 monomers for both systems, see Fig. 1c) increases due to the induced disassembly of the larger aggregates. However, whereas in the non-cooperative **M** system we observe a generalised and more homogeneous impact, with a global shift of the size distribution toward smaller sizes (see also Fig. S4), in the **M$_{coop}$** system the assemblies larger than the average size-range (>17–32) suffer less the stimulus (notice the inversion between the two systems for sizes 33–64 in Fig. 3e). The impact of **C**

cappers is absorbed by the free monomers, which are already present in solution (typical of the cooperative systems) and which increase considerably in number due to depolymerisation of larger aggregates.

The data demonstrate that two systems with a very similar degree of polymerisation but different nature of inter-monomer interaction (in terms of cooperativity) can produce a different response to perturbation, resulting from the aggregate-size distributions that they generate at equilibrium. In particular, the cooperative system is populated by a large fraction of free monomers and by strong and more stable long fibres. Thus, in the **M$_{coop}$** system the smaller units absorb large part of the stimulus, preserving the longer fibres from perturbation. This is not observed in the non-cooperative (isodesmic) system **M**, where the assemblies suffer the perturbation in a more uniform way. Moreover, the perturbation reinforces the intrinsic differences that are already present in the unperturbed systems (see Fig. 2b). In fact, upon perturbation, the **M** system preserves the unimodal size distribution of a non-cooperative system, while in the **M$_{coop}$** system the bimodal size-distribution is even more emphasized, as the concentration of free monomers increases.

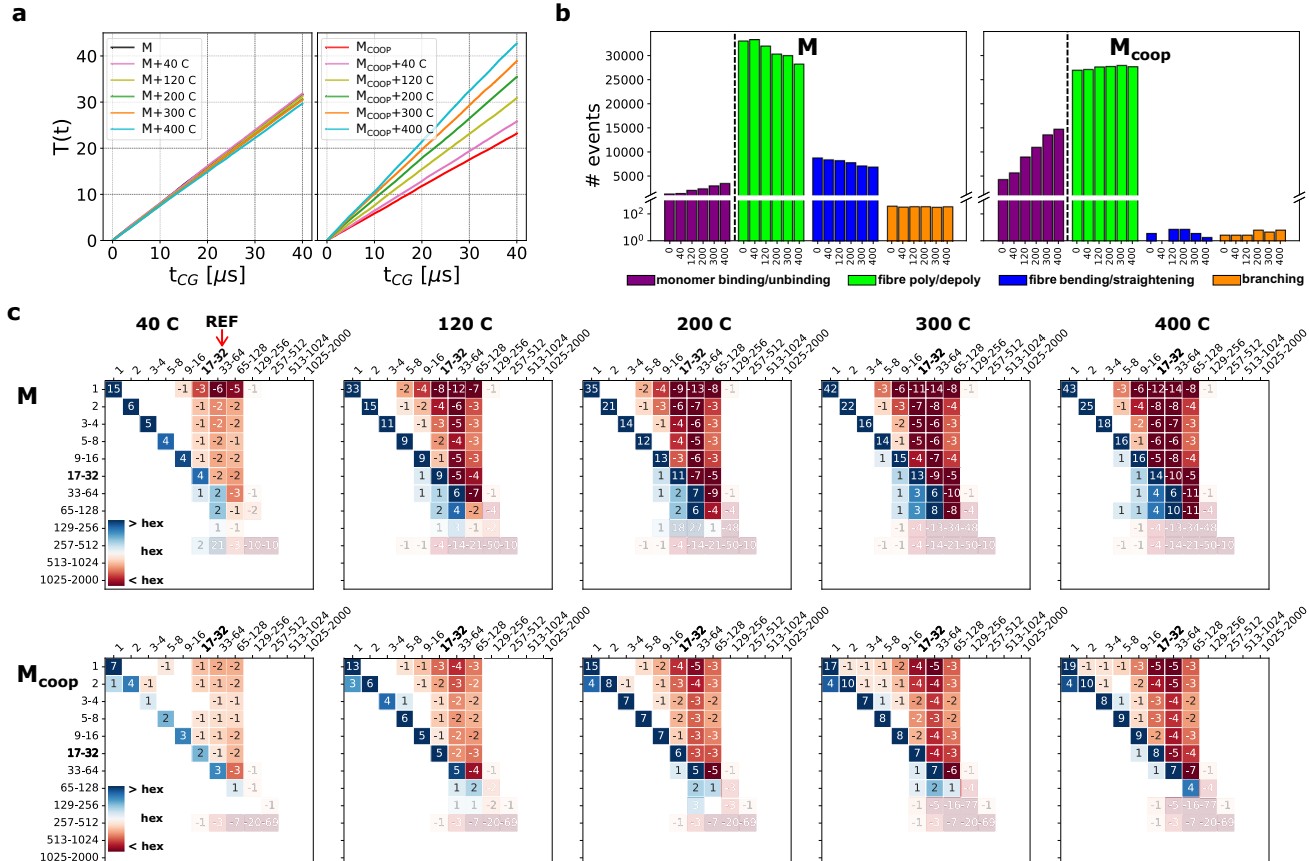

**Fig. 4 | Impact of perturbation of M and M_coop systems dynamics. a** Molecular traffic $T(t)$ for the **M** and **M_coop** systems in equilibrium with different amounts of **C**. **b** Dynamic events detected by LENS[43] for the **M** and **M_coop** systems in equilibrium with different amounts of **C**. See Fig. 2d for the classification of events and Fig. S3 for the full analysis. **c** Δ probability (in %) matrices of **M** (top) and **M_coop** (bottom)

systems (with $\Delta t = 15$ ns), as the amount of chain-stoppers is increased (higher perturbation strength). The size-range including the average sizes of the unperturbed systems is highlighted in bold text. The matrix areas of events involving size-ranges above 65–128 are shaded as they are based on insufficient statistics.

## Collective stimuli-responsive behaviours emerging from microscopic dynamics

We investigated how the molecular communication dynamics, which differs between the two systems (Fig. 2), changes due to the perturbation induced by **C** addition. First, the molecular traffic changes in a strikingly different way: the addition of **C** mildly reduces the average dynamicity of the **M** system (the traffic decreases by 5% at the highest **C** concentration). Conversely, the dynamicity of **M_coop** increases significantly, as the traffic almost doubles at the largest **C** concentration. We used a LENS[43] analysis to investigate which communication mechanisms determine such different traffic variations. As shown in (Fig. 4b), in the **M** system the decrease of traffic is driven by a drop in the number of multimer polymerisation-depolymerisation events (in green). On the other hand, in the **M_coop** system the traffic increase is mostly due to monomer exchange events, which triplicate as the **C** concentration increases.

We also calculated how the transition probability matrices in Fig. 2b -providing a detailed microscopic-level picture of the communication network between assemblies- change upon chain-stopper addition. In Fig. 4c we show Δ matrices, displaying the changes of equilibrium transition probabilities experienced by the two systems as the **C** content increases ($\Delta P_{ij} = P_{ij}^{(perturbed)} - P_{ij}^{(unperturbed)}$). Each Δ matrix entry reports the variation in terms of probability of a specific transition event in the perturbed system with respect to the unperturbed one. Positive values (blue shades) indicate increased probabilities, while negative values (red shades) indicate decreased probabilities.

These matrices showcase interesting tendencies of the two systems, determined by the increase of **C** content. We first underline the

trend detected for polymerisation events (upper-triangular matrices). The polymerisation probabilities are impaired due to the addition of **C** monomers in both **M** and **M_coop** systems. Binding to the tips, the chain-stoppers inhibit the transitions that feed the growth of large-size aggregates, in agreement with the shift of size distributions to smaller sizes (Fig. 3d). Nonetheless, it is clear that the dominant fibre sizes are more impacted (for what concerns the natural tendency to polymerise toward larger sizes) in the non-cooperative **M** system (see dark red columns for size-ranges 17-32 and 65-128, Fig. 4c) than in the **M_coop**.

Even more striking differences between the two systems emerge when considering the impact of the perturbation on depolymerisation events (lower-triangular matrices). While in the **M** case depolymerisation probabilities increase evidently for large assemblies (>17–32, in cyan in Fig. 4c), this does not happen in the **M_coop** system (except for very long fibres, that are insufficiently sampled in the simulations). In the **M_coop** system we notice a marked increase in the depolimerisation probability of the assemblies that are smaller than the critical size (i.e., dimers), which become very unstable and gain further importance as source of free monomers.

All these results underline how the different ensemble response of non-cooperative **M** and cooperative **M_coop** systems is related to their diversity at the level of structure (mono-modal vs bi-modal size-distribution) and microscopic dynamics: while in both systems the chain-stopper insertion hampers polymerisation, inducing the reduction of aggregate sizes, in the **M** system the **C** insertion reduces the average dynamics (traffic), enhancing the depolymerisation of large size aggregates, with the result of an indiscriminate shift of size distribution toward smaller aggregates. On the other hand, in the

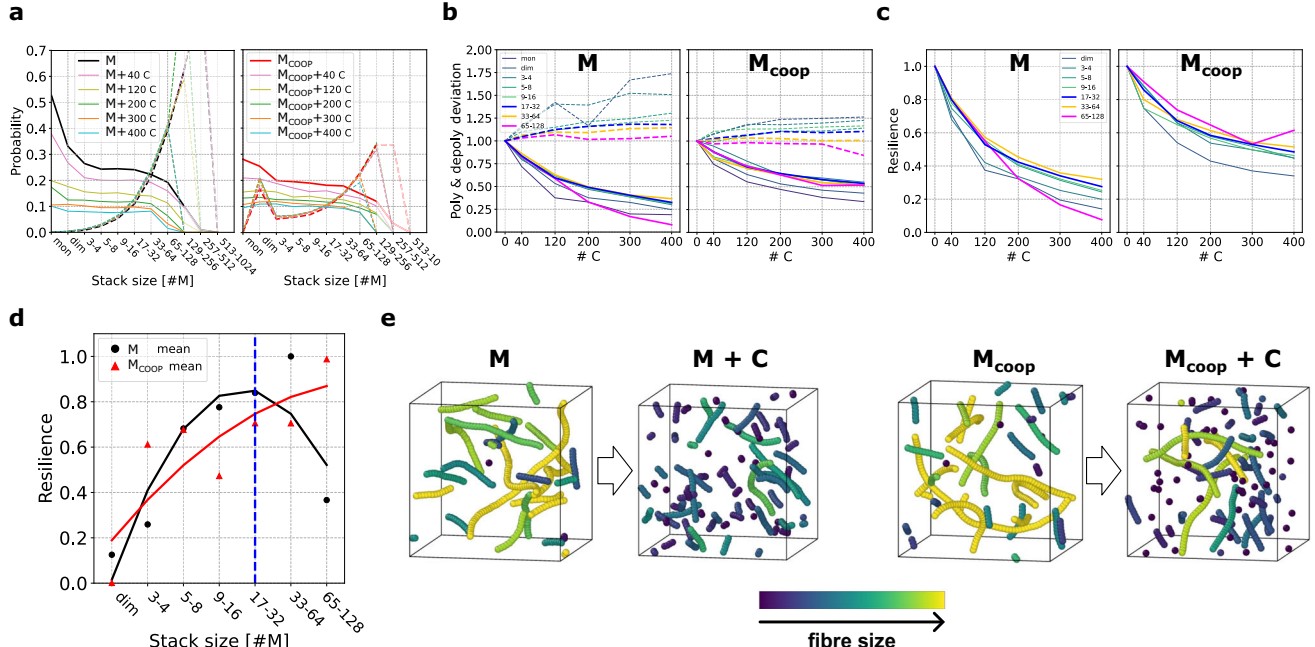

**Fig. 5 | Resilience of M and M$_{coop}$ systems upon chain-stopper insertion. a** $P_{poly}$ (solid) and $P_{depoly}$ (dashed) (with $\Delta\tau = 15$ ns) as a function of the aggregate size, for **M** and **M$_{coop}$** systems in equilibrium with different amounts of **C**. For large aggregates (size > 128) insufficient statistics provides inaccurate estimates of transition probabilities (the presence of such large aggregates is rare in the unperturbed systems, and they tend to disappear as the concentration of **C** increases). Poor statistics is indicated by shaded colours. **b** Relative probabilities of polymerisation (solid) and depolymerisation (dashed) of assemblies of different size-ranges, as a function of the **C** content (see "Methods" for details), for **M** and **M$_{coop}$** systems. **c** *Resilience* (see text for the definition) of assemblies of different size-ranges, as a function of the **C** content, for **M** and **M$_{coop}$** systems. **d** Normalised resilience of **M** (black circles) and **M$_{coop}$** (red triangles) systems averaged over the strength of the perturbation (number of **C**), per each size range. The values are normalised so that the minimum is 0 and the maximum is 1, and quadratic fits (solid lines) are reported to highlight the different trends exhibited by the two systems. The blue dashed line indicates the average aggregate size in the unperturbed systems. **e** Equilibrium snapshots of unperturbed/perturbed **M** (left) and **M$_{coop}$** (right) systems (500 monomers in equilibrium with 0 or 100 **C**). Different aggregates are coloured according to their size (see Supplementary Movie 1 for inter-assembly dynamics).

**M$_{coop}$** system, the dynamic network reacts differently: indeed, the addition of **C** cappers increases the traffic, mainly due to the increase of monomeric and dimeric entities (below the critical size, which are unstable). In this way, in the cooperative system, the impact of the perturbation is absorbed by the smaller (and weaker) assemblies, which allows the **M$_{coop}$** system to better preserve the larger (stronger) assemblies as compared to the **M** system. These data demonstrate how cooperativity determines a size-selective response to the chemical perturbation. This is not simply due to structural factors— i.e., to the large concentration of free monomers that capture the **C** cappers—but rather to microscopic diversity of the dynamical exchange network that interconnects the different aggregation states. The fact that in a cooperative system, the smaller/weaker aggregates absorb the perturbation, preserving the larger/stronger assemblies, indicates that the native, unperturbed differences between weaker and stronger assemblies are amplified further by the perturbation.

Stimuli-responsiveness emerges from the internal microscopic dynamics underpinning the systems, i.e., from the "local features" of the network of molecular exchange between all the different-size aggregates at the equilibrium. At the same time, this "resilience" of the fibres is not simply a property of the individual monomers or assemblies, but rather a collective property emerging from the network of communications that connects all the entities that populate the system. For this reason, understanding the origin of such collective stimuli-responsive behaviours, and of the mechanisms underpinning their amplification, requires a microscopic-level analysis of these complex systems (as standard ensemble averages alone do not capture where such behaviours originate from, nor the processes controlling them).

## Network resilience to external perturbations in complex molecular systems

To attain a more quantitative characterisation of the diverse microscopic internal response of **M** and **M$_{coop}$** systems to the addition of chain-stoppers, we computed the polymerisation and depolymerisation probabilities associated to assemblies of different sizes, and assessed how these change as a function of the amount of **C**s introduced (Fig. 5a). The data summarise the transition matrices of Fig. 4, highlighting how the addition of the chain-stoppers sensibly reduces polymerisation in both systems while affecting depolymerisation in a qualitatively different way.

We can quantify how the intensity of the stimulus (the **C** concentration) affects the internal dynamics of exchange between aggregates, by evaluating how the probability of polymerisation/ depolymerisation of all the different-size entities that populate the systems change under perturbation with respect to the unperturbed equilibrium conditions. These relative probabilities of polymerisation/ depolymerisation shift away from 1 (reference/unperturbed value) as the stimulus impacts the native dynamics of a specific assembly size (or size-range). The trend of these relative probabilities (Fig. 5b) clearly shows how the impact of perturbation is less relevant in the **M$_{coop}$** system (right) than in the **M** system (left) both in polymerisation and depolymerisation. This is particularly evident for larger size aggregates, which are relatively less perturbed in the **M$_{coop}$** system.

To quantify the perturbation impact, we define the microscopic *resilience*, as the ratio between the relative polymerisation and depolymerisation probabilities, for all assembly sizes, as a function of the **C** content (see "Methods" for further details on the definition). Akin to an equilibrium constant variation, this quantity estimates how much the perturbation affects the dynamic equilibrium (stability) of the different

assemblies. The resilience curves (Fig. 5c) show that this quantity is overall larger in the $M_{coop}$ system, remaining closer to the unperturbed value, compared to the $M$ system. Furthermore, resilience follows different trends across size ranges between the cooperative and non-cooperative systems. As shown in Fig. 5d, in the $M$ case, the resilience increases with size until a maximum at size-range 17–32, after which the value falls. This shows how in a non-cooperative system, the resilience of the assemblies simply follows the size distribution (i.e., the most probable/favoured size having higher resilience than the smaller and longer ones). On the contrary, the $M_{coop}$ system shows a different behaviour. The resilience keeps increasing with the size of the constructs, and assemblies larger than the most favoured ones (e.g., 65–128) have a higher resilience.

The non-obvious system's ability to preserve the resilience of assemblies larger than the average is due to the cooperativity of interactions, and specifically to the non-uniformities that this feature introduces in the microscopic dynamics. As shown, the stimulus affects the unstable entities smaller than the critical nucleus of aggregation. This preserves the integrity of larger entities that remain stable (see equilibrium unperturbed and perturbed configurations in Fig. 5e).

The difference in the ensemble behaviour of cooperative vs. non-cooperative systems lies in the concept of *super-additivity*. In a cooperative system, interactions are super-additive, promoting the growth of larger aggregates and unfavouring aggregates smaller than a critical size (see also free-energy profiles in Figs. S11, 12, associated to polymerisation equilibrium of the two systems). Let us assume that in a self-assembling system intermonomer interactions are a resource (such that aggregation is a resource gain) that the system optimizes reaching equilibrium. Under this assumption, a cooperative system shows resource super-additivity, where entities that grow will tend to acquire more resources at the expense of smaller entities. This results in a non-uniform behaviour, with a critical nucleus below which smaller unstable entities are poor in resources, and above which larger entities grow and become richer in resources. When such a system is perturbed by subtracting resources (e.g., adding inhibitors as in this case, raising the temperature, etc.) the ensemble compensates by targeting the stimulus towards the smaller, weaker units, thus preserving the stronger entities. In a non-cooperative system, without resource super-additivity, there are no weaker or stronger groups, but a unimodal distribution with a certain favoured size that essentially depends on the concentration of units. The resilience follows such size-distribution, and the system responds homogeneously. Moreover, analogous behaviour emerges in smaller-scale systems containing 500 monomers when perturbed by the chain-stopper stimulus, confirming that our results are not vitiated by finite-size effects (see Figs. S8, S9) and, globally, the robustness and generality of our conclusions.

Interestingly, these results demonstrate how behaviours typical of more complex, large-scale systems (e.g., economic, social, etc.) may emerge also in relatively simple (both in terms of structure of the units and number of species) self-organising systems, such as those studied herein. In the considered supramolecular systems, composed by a single, structurally simple species, complex behaviours may emerge from the intrinsic complexity of their internal dynamics, which originate from structural and dynamical non-uniformities induced by additivity/cooperativity of monomer-monomer interactions.

## Discussion

In this work, we investigate how supramolecular polymer systems—which even in the simplest system variants possess typical features of complex molecular-scale systems—react to perturbations induced by external chemical stimuli. In particular, we demonstrate how stimuli-responsiveness is an emergent ensemble property that originates from the microscopic-level dynamical complexity of these systems.

To explore how a supramolecular system that possesses innate internal dynamics responds to an external stimulus, we employed minimalistic molecular models allowing us to study large numbers of interacting monomers that self-assemble into populations of differently-sized aggregates (fibres). As a representative case study of external stimulus, we considered supramolecular polymer systems perturbed *via* the addition of molecular cappers ($C$): asymmetric terminal monomers that, once binding to the tips of fibres, inhibit their growth, perturbing the native polymerisation/depolymerisation equilibria and inducing an ensemble response[8]. We developed minimalistic CG models of two prototypical systems: $M_{coop}$, in which the monomers self-assemble in a cooperative fashion, and $M$, where the monomers polymerise in a isodesmic non-cooperative process. The two models have been designed to exhibit comparable average equilibrium properties: e.g., similar propensity to self-assembly and average fibre size (~26–27 monomers, see Fig. 1). However, they show different microscopic features, such as, e.g., the monomodal vs. bimodal equilibrium aggregate size-distributions, formed respectively by the non-cooperative ($M$) and cooperative ($M_{coop}$) systems, where the latter shows the critical size/nucleus of polymerisation (~3–4 monomers, see Figs. 1c, 2b–d) implying coexistence of free monomers and longer fibres in equilibrium (see also Supplementary Movie 1).

At the equilibrium, the two systems are composed of a variety of assemblies that communicate *via* monomers/oligomers exchange generating complex dynamical networks (Fig. 1e). This suggests a paradigm shift, where stimuli-responsiveness is not closely related to the response of the individual assemblies, but it is rather an emergent, collective property, related to how such dynamical exchange networks vary under perturbation. Tracking all the monomer transitions across different-size aggregates at every simulation time-step we thus studied such supramolecular systems as complex systems[34]: i.e., the subject becomes the inherent dynamical network (characterised in Fig. 2), and how it changes when perturbed (Fig. 3). Such an approach is highly informative as it allows to reconstruct the internal complexity (in terms of structure and dynamics) of the system. From the resulting microscopic-level information one can indeed estimate *a posteriori* macroscopic (average) properties, but also understand the factors determining them[34,43–46] (while the opposite, going from average macroscopic estimations to microscopic information, is typically prohibitive).

The binding of chain-stoppers ($C$) inhibits the growth of the fibres, disturbing the equilibrium of the systems and impacting their structural and dynamic properties. Our studies show that the global response of the cooperative and non-cooperative systems to the stimulus is similar (Fig. 3d). Identifying monomer binding events as resources, the data indicate how the the addition of the $C$ species, inhibiting fibre growth, subtracts such resources to the systems. Figure 3d shows how, as both systems undergo the a similar resources' deprivation, the effect of the perturbation is macroscopically the same in the ($M_{coop}$ and $M$) systems.

However, at the same time, the systems respond in a radically different manner on a microscopic level. Detailed analyses of how the ($M_{coop}$ and $M$) systems respond in terms of structure and dynamics of all the assemblies populating them (Figs. 3, 4, and 5) demonstrate that in the non-cooperative system all assemblies suffer from the stimulus in a similar way. Under perturbation, the $M$ system preserves the monomodal assembly-size distribution (shifting to smaller sizes). Instead, in the cooperative system, the impact of chain-stoppers is size-selective and perturbs different assemblies in different ways. In this case, the perturbation accentuates the bimodality of the distribution, and the stronger/longer assemblies are better preserved at the expense of the smaller/weaker entities (see data in Fig. 5 and Supplementary Movie 1).

The minimalistic representation provided by the models used herein allows us to observe how the dynamical complexity of these

supramolecular polymers' systems, and the responsive behaviours that emerge from them are tightly dependent of the cooperativity in the monomer-monomer interactions. In general, the results indicate how in a cooperative system, when the system undergoes stress, the larger, more stable entities become stronger as compared to the weaker smaller assemblies, while in a non-cooperative system the internal response to the perturbation is more homogeneous. Noteworthy, such different microscopic-level ways of responding to an external perturbation simply reflect the best (thermodynamic) ways for the two systems to evolve upon an external perturbation and to optimize the resources (the interactions). Qualitatively, this provides a relevant insight for supramolecular polymer systems in general: the higher is the cooperativity in the self-assembly, the more bimodal will be the distribution of the assemblies (in terms of size and relative stability)[47–49], and the higher will be the resilience of larger entities as compared to smaller ones.

Intriguingly, these results demonstrate how collective behaviours typical of more complex, higher-scale systems (e.g., social, economical, environmental, etc.) may emerge already in very simple self-assembling systems. The models studied herein are among the simplest possible ones (mono-species supramolecular homo-polymer systems, composed of elementary interacting units). Nonetheless, collective stimuli-responsive behaviours emerge from their internal (microscopic) dynamic interaction network, which can be very complex and microscopically diverse even in the simplest self-assembling system.

This study offers useful approaches to track the emergence and amplification of collective behaviours, and to learn how to encode (at the molecular level) them in artificial self-assembling systems with programmable emergent functionalities. Specifically, the results obtained by our models indicate how the design of monomers that can interact and self-assemble cooperatively in a controllable way may offer the opportunity to control collective, emergent properties, for example determining the resilience to the perturbation induced by chain-stoppers. Moreover, our work offers a simplified platform to study the mechanisms underpinning the emergence and amplification of collective behaviours in complex molecular and supramolecular systems, thought of as smaller-scale analogs of higher-scale (micro- and macro-scopic) complex systems.

## Methods

### Minimalistic models

As representative of non-cooperative polymerising system we employed the **M** monomers studied in ref. 34. The **M** monomer is composed of seven beads in total, six shielding beads at the vertices of a hexagon and a central core bead (Fig. 1a), bound together by harmonic bonds. Specifically, the force constant and the equilibrium length between the nearest neighbour beads are 20,000 kJ mol$^{-1}$ nm$^{-2}$ and 0.47 nm respectively. Each shielding bead is also connected with the one at the opposite vertex of the hexagon by a harmonic bond with force constant and equilibrium length of 15,000 kJ mol$^{-1}$ nm$^{-2}$ and 0.94 nm, respectively so that the molecules are maintained planar. The non-bonded interactions between central beads (red) are defined by LJ potentials, with constant $\sigma = 0.47$ nm and interaction strength $\epsilon = 45$ kJ mol$^{-1}$ (Fig. 1a).

The **M**$_{coop}$ model has the same hexagonal topology and bonded interactions of the **M** model, with the addition of a dipole centered on the central monomer bead composed of two charged beads ($q = \pm 1.4\,e$) constrained at fixed distance from each other of $d = 0.28$ nm (each placed 0.14 nm from the center of the core bead). The two beads are kept aligned along a straight vector (see Fig. 1b) via a harmonic angular potential that maintains an angle of 180° between the charged beads and the core bead (with force constant of 1500 kJ mol$^{-1}$). Such dipole charged beads have mass 1/3 of the mass of the other (regular) beads and interact only via Coulomb interactions (no van der Waals

interactions). In this way, charged beads do not bear excluded volume, forming a dipole that is free to rotate around the center of the molecule (without any interaction with the other monomers beads) and that can interact only with the dipoles of the other monomers. The addition of dipoles implies an entropic penalty for the binding of two free **M**$_{coop}$ monomers, having randomly oriented dipoles, as compared to the binding of **M**$_{coop}$ monomers (or fibres) to a pre-assembled fibre, where the dipole vectors at the ends are optimally oriented to favour attraction. Therefore, as discussed in the "Results" section, rotating dipoles enhance monomer-monomer interaction directionality and offer additional degrees of freedom that favour the formation of fibres larger than a critical nucleus, thus imparting cooperativity. No long-range electrostatics are included in the simulations in order to preserve the local character of the dipole-dipole interactions and to avoid spurious long-range attractions/orientations in the **M**$_{coop}$ model.

The chain-stopper **C** model is a variation of the **M** monomer, with an additional shielding bead constrained by harmonic bonds and placed out of the molecule plane, along its axis, adjacent to the core bead (see Fig. 3a). This extra bead acts as excluded volume (it weakly interacts with all the other beads *via* $\epsilon = 0.2$ kJ mol$^{-1}$) preventing the supramolecular binding of other monomers along its direction. The chain-stoppers **C** inherit the forcefield from the **M** monomers, thus interacting mainly *via* LJ potential with constant $\sigma = 0.47$ nm and interaction strength $\epsilon = 45$ kJ mol$^{-1}$, for both the self (**C-C**) and the cross (**C-M,C-M**$_{coop}$) interactions. The chain-stoppers interact with the **M**$_{coop}$ monomers only *via* LJ potential (core beads, $\sigma = 0.47$ nm and $\epsilon = 45$ kJ mol$^{-1}$), unaffected by the dipole charges. We underline that in all the reported analyses of aggregation the **C** monomers are not explicitly counted as part of the assemblies (i.e., if a **C** monomer is bound to a **M** one, the **M** is considered free and has coordination $\phi_i = 0$). Thus, the **C** species is accounted only for the effect that it imparts on the structure/dynamics of the **M** or **M**$_{coop}$ monomers. For supporting analyses, we also performed simulations of "purely repulsive" variants of **M** or **M**$_{coop}$ models, that we name **M**$^R$ or **M**$^R_{coop}$, respectively. These systems mirror the **M** or **M**$_{coop}$ models, except for the fact that attractive interactions have been removed, by truncating and shifting the LJ potentials at the minimum ($r = 2^{1/6}\sigma$) and by setting the dipole charges to 0. The simulation of **M**$^R$ or **M**$^R_{coop}$ is used to compare the dynamics of free monomers to that of the interacting systems at equilibrium (see Fig. S10).

### CG-MD simulations

The CG-MD simulations of the **M** and **M**$_{coop}$ are performed by the GROMACS software[50] (version 2021.4) in NVT conditions, using a constant volume for the simulation box of $35.012 \times 35.012 \times 35.012$ nm$^3$, with periodic boundary conditions and a constant temperature of $T = 300$ K. The number of molecules is kept constant to $N = 2000$, which consists in a monomer density of 0.0466 nm$^{-3}$, corresponding to a concentration of ~77 mM. A variable number of **C** monomers is added in both the **M** and **M**$_{coop}$ systems (0, 40, 120, 200, 300, 400), obtaining a total of $2 \times 6$ sets of simulations. An additional set of simulation of a smaller model system, containing $N = 500$ monomers (in a cubic simulation box of volume $22.056 \times 22.056 \times 22.056$ nm$^3$), having the same concentration of monomers and of the larger ($N = 2000$) systems, has been carried out. The dynamical equilibrium of such smaller **M** and **M**$_{coop}$ variants has been analyzed as done for the larger systems. When perturbing these smaller model systems, the same capper concentrations as in the larger analogs was maintained (adding 10, 30, 50, 75 and 100 **C**s). These models demonstrated consistent behaviours as those seen in the larger $N = 2000$ systems (see Figs. S7–S9).

To simulate dynamics, implicit-solvent Langevin equations are solved for all the beads, in which the stochastic term implements both the solvent friction and thermal fluctuations of the system. The stochastic dynamics (sd) integrator is used with a time step of $\Delta t = 20$ fs; the inverse of the friction constant (tau-t) set to 0.1 ps, which also

sets the coupling with the random force term governing the system's temperature. The non-bonded interaction potentials are truncated at a distance of $r_c = 1.1$ nm and shifted to zero.

For the **M** and **M**$_{coop}$ systems without chain-stoppers, we performed a total of 60 μs of simulation starting from uniformly distributed monomers. Both systems reach the self-assembled equilibrium state in the μs scale (see also Fig. S1 and ref. 34), therefore we analyzed the last 40 μs to assess the equilibrium behaviour. The **C** monomers are added randomly to the last equilibrium snapshots of the unperturbed simulations, and then simulations of 120 μs are performed, again keeping the last 40 μs as the equilibrium part of the trajectory (see Fig. 3b). We notice that the additional **C** monomers equilibrate within the systems over ~40 μs (see Fig. S5), this motivates the longer simulation times. After reaching the established equilibration time, we then studied the supramolecular equilibrium dynamics by prolonging the CG-MD of each of the 12 systems (**M** and **M**$_{coop}$ systems with variable amount of **C** monomers) for 40 μs, where the conformations are sampled every 0.3 ns. Further CG-MD simulations were performed for the **M** and **M**$_{coop}$ models, to assess the average diffusion and rotation dynamics of the monomers at equilibrium. Additional 5 μs of MD were run starting from the final equilibrium configuration of the production runs. A faster sampling time of 50 ps was adopted here, to have a better detail on the dynamics. Similarly, 5 μs of MD were performed for the "purely repulsive" **M**$^R$ and **M**$^R_{coop}$, following an equilibration stage of 2500 ps (sufficient for non-attractive monomers). For the latter system we also performed 50 ns of MD with a 0.5 ps sampling time, to fully capture the fast rotation dynamics of the free dipoles

## Analysis of the CG-MD simulations

All the analyses are performed using Python scripts employing the MD-Analysis package[51,52]. The rendered snapshots in Figs. 1, 2, 3, 5 are generated using VMD[53] and Ovito[54].

For the monomer aggregation analyses, we developed an improved version of the analysis reported in ref. 34. Instead of considering that two **M** monomers belong to the same assembly when they have a core-core distance below a certain cutoff distance, the new analysis uses two cutoffs, a binding cutoff $r_b$, and an unbinding cutoff $r_u$, where $r_b < r_u$. Two monomers that are not bound together at a certain time $t$ form a bond if their core-core distance goes below $r_b$ at the next instant $t + \Delta t$, where $\Delta t$ is the sampling step employed for the analysis of the dynamics. Instead, when two monomers are already bound at time $t$, their bond breaks only if their distance exceeds $r_u$ at the following sampling step $t + \Delta t$. In this way, we suppress possible fluctuations in which bonds break only momentarily (i.e., for $\Delta t$) and are reestablished right away. In the following analyses, we set $r_b = 6$ Å and $r_u = 9$ Å.

To analyze binding/unbinding events involving each monomer during the equilibrium phase of CG-MD with a sampling step of $\Delta t = 0.3$ ns, we compute transition matrices (see Fig. S2). The entries of the matrices record the number of monomer transitions from an assembly of size $i$ (row index) to an assembly of size $j$ (column index); Due to detailed balance these transition matrices are symmetric. By normalising the transition matrices so that each row sums to 1 we obtain transition probability matrices $T(\Delta t)$, indicating the probabilities that a monomer in an assembly of size $i$ transits to an assembly of size-range $j$ during the interval $\Delta t$. A transition probability matrix can be multiplied by itself extending the timespan of the probabilities:

$$T(n\Delta t) = [T(\Delta t)]^n, \qquad (1)$$

We use this relation to compute transition probabilities over a time interval $\Delta \tau = 15$ ns. To provide more readable and less noisy probability values, the entries of the matrices are grouped into binary size-ranges.

The polymerisation and depolymerisation probabilities $P_{poly}$ and $P_{depoly}$ as a function of the aggregate size (Figs. 2c and 5a), are computed from the transition probabilities computed over a $\Delta \tau = 15$ ns. $P_{poly}$ sums all probabilities associated to transitions that increase the aggregate size, whereas $P_{depoly}$ sums all probabilities associated to transitions that decrease the aggregate size. For clarity of visualisation and to suppress noise, in Figs. 2c and 5a the polymerisation and depolymerisation curves are CG into binary size-ranges.

From $P_{poly}$ and $P_{depoly}$ we can quantify the response of the system dynamics to the addition of **C**s. We measure how the polymerisation and depolymerisation probability associated to a specific size-range change as a function of the concentration of **C**s, obtaining the $P_{poly}([C])$ and $P_{depoly}([C])$ shown in Fig. 5a. We then define the relative polymerisation (or depolymerisation), namely the ratio between the perturbed and unperturbed probabilities of polymerisation $P_{poly}([C])/P_{poly}(0)$ (or depolymerisation $P_{depoly}([C])/P_{depoly}(0)$). The relative probabilities associated to different size ranges are reported in Fig. 5b. Gathering these quantities we then define the *resilience*, as the ratio between the polymerisation and the depolymerisation deviations. The resilience is given by:

$$resilience = \frac{P_{poly}([C])P_{depoly}(0)}{P_{depoly}([C])P_{poly}(0)} = \frac{R_{equil}([C])}{R_{equil}(0)}. \qquad (2)$$

The ratio $R_{equil} = P_{poly}/P_{depoly}$ is a measure of the balance between the polymerisation and depolymerisation propensity that regulates the dynamics of a certain aggregate size (or size range). Therefore, the *resilience* indicates how this balance changes when chain-stoppers are added, relatively to the unperturbed system. Resilience is thus a measure of how aggregates are impacted by the perturbations.

The diffusivity of monomers in the **M** and **M**$_{coop}$, and in the purely repulsive variants **M**$^R$ and **M**$^R_{coop}$ is obtained from the average mean-squared-displacement (MSD) of all monomers' centers of mass (Fig. S10a, b) as a function of the displacement time $\tau$. The diffusion constant $D$ is then obtained from the fitted angular coefficient of the MSD via Einstein's relation (MSD$(\tau) = 6D\tau$). Finally, rotational auto-correlation functions of the monomers are computed by associating to each monomer a vector perpendicular to its plane and evaluating the vector autocorrelation function in time. Same procedure was followed for the dipole vectors in the **M**$_{coop}$ (and **M**$^R_{coop}$) systems.

**LENS analysis.** To inquire about the communication mechanisms, we employ the LENS (Local Environment and Neighbours Shuffling) descriptor[43], which quantifies the local reshuffling around chosen classes of particles by counting the number of changes in a defined neighbourhood, along a time interval $\Delta t$ (in this case $\Delta t = 0.3$ ns). We consider the monomer cores as labelled particles of this analysis, in such a way that the detected neighbourhood transitions follow the supramolecular communication dynamics, see ref. 43 for details. As representatives of the equilibrium part of the trajectories, we analyzed the last 5 μs of each simulation. Defining the neighbourhood by using a cutoff radius of 7 Å we obtain the LENS time series for the perturbed and unperturbed systems, from which it is possible to identify six different discrete values, instead of a distribution of LENS values which is the typical outcome of a LENS analysis in systems characterised by diverse level of dynamicity (see ref. 43 for details). Such discreteness of the LENS values is due to the triviality of the reshuffling that each particle can have when in a fibre, which is almost always constrained between two other fixed particles unless the fibre breaks or interacts with another fibre in that specific area. The same applies also to monomer, which does not have any reshuffling, not having neighbour particles around unless they interact with a fibre/monomer. Figure S6 shows, as an example, the LENS time series for the unperturbed **M** and **M**$_{coop}$ systems. Each LENS value corresponds to a different molecular transition event: in

gray are coloured the bare fibre monomers that do not change their neighbourhood, in purple a single monomer binding/unbinding to another one (the latter being either part of an aggregate or not), green identifies polymerisation and depolymerisation of fibres (of size ≥2), blue identifies non-neighbouring monomers of the same fibre that either form or break a contact (fibre bending/straightening), orange is the formation (or elimination) of a branch, which is usually not stable, red is a rare and ephemeral event of branching (and un-branching) involving four centers. In Figs. 2e and 4b are reported the four most relevant mechanisms, coloured accordingly, while the whole LENS analyses with all the mechanisms are reported in the Supplementary Information (Fig. S3).

**Well-Tempered metadynamics calculations.** The free-energy profiles associated to the dimerisation in the **M** and **M**$_{coop}$ systems have been evaluated via Well-Tempered Metadynamics (WTMetaD)[41] simulations, that enhance the sampling of the system conformations allowing the accurate estimation of ensemble averages in reduced simulation times. We simulated systems containing 2 **M** (or **M**$_{coop}$) monomers in a box of size ($3.5012 \times 3.5012 \times 3.5012$ nm$^3$), having the same concentration of the unperturbed $N = 2000$ monomer systems. The WT-MetaD history-dependent bias potential was applied on the distance between the monomer centers, stimulating the binding and unbinding[35–37,40]. The biasing potential was built by depositing every 10 ps Gaussian kernels of width equal to 0.1 nm and initial height of 0.1 kJ mol$^{-1}$. A bias factor of 5 was chosen to regulate the decrease in the kernel height, and the convergence of the deposited bias. In 1 µs of WT-MetaD, both systems efficiently sampled binding/unbinding events, converging to the dimerisation free-energy profiles shown in Fig. S12a. As shown in refs. 55–57. we employed WT-MetaD calculations to estimate the free-energy ($\Delta F$) and internal energy profiles ($\Delta U$) as a function of the distance between the monomer centers. By subtracting the internal energy from the free-energy profile we then obtained the entropic ($-T\Delta S$) term (see Fig. S12b, c). Finally, in the case of the **M**$_{coop}$ model, we also estimated the free-energy difference associated to fibre elongation by exploring with WT-MetaD the binding/unbinding between two monomers, one of which has its dipole vector restrained (via harmonic angular potentials) along the direction orthogonal to the monomer plane, as it would be the case for a monomer located at the tip of a fibre. The result, in Fig. S12d, shows that the free-energy gain associated to fibre growth via monomer addition is larger than the gain determined by the association of two disassembled monomers, supporting the cooperativity of the self-assembly in the **M**$_{coop}$ model.

## Data availability

Additional details and simulation data are provided in the Supplementary Information. Complete data on the molecular models, the MD simulations and the analyses performed in this work have been deposited at https://doi.org/10.5281/zenodo.15488369. Other related data are available from the corresponding authors upon request.

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

## Acknowledgements

G.M.P. acknowledges the support received by the European Research Council (ERC) under the European Union's Horizon 2020 research and innovation program (Grant Agreement no. 818776—DYNAPOL). The authors also acknowledge the computational resources provided by the Swiss National Supercomputing Center (CSCS).

## Author contributions

G.M.P. conceived this research. M.C. and C.P. developed the molecular models and performed the simulations. M.C., C.P., and G.M.P. analyzed the results. G.M.P. and C.P. supervised the work. All authors wrote the manuscript.

## Competing interests

The authors declare no competing interests.
