## [Peer Review file · Nature Communications]

Non-Trivial Stimuli-Responsive Collective Behaviours Emerging from Microscopic Dynamic Complexity in Supramolecular Polymer Systems

Corresponding Author: Professor Giovanni Pavan

Version 0:

Reviewer comments:

Reviewer #1

(Remarks to the Author)

This manuscript describes the study of a dynamic network models where small molecules are able to assemble into large fibers due to attractive physical interactions. When introducing a small fluctuating dipole to the assembling agents, the dynamics are altered, promoting the aggregation in a similar structure, but with a bimodal distribution comprising a large number of highly dynamic units that dynamically interact with the rest of the network. While the emerging behavior of the cooperative and non-cooperative systems are quite similar, their response to external stimuli (here triggered by introducing capping agents) is very different. While this stimulus uniformly triggers a reduction of filament size in the non-cooperative system, it weakly affects the cooperative system. The reason is that the large number of small dynamic unit can shield the large filament against the action of capping agents. The authors therefore argue that the idea of cooperation is important for system resilience against changes in external environment.

Generally, the paper is well-written, and results are presented in a convincing manner. The scientific approach is sound and nicely conducted. The comments below are mostly about the relevance of this work to either polymeric systems, or general complex systems.

1. Does the network under study represent any real physical system? Has the type of cooperativity observed in the simulations been reported before, either in natural or synthetic systems? Or is this primarily a toy model aimed at exploring the concept of cooperativity? Providing more context about the system's relevance to real-world scenarios would strengthen the impact of the study.

2. While the authors convincingly argue that cooperativity enhances system resilience (as defined in their framework), it is not immediately clear how broadly this concept applies. Could there be cases where cooperativity might reduce resilience? Discussing potential counterexamples or boundary conditions for this concept would provide a more comprehensive understanding of the phenomenon.

3. The study offers interesting insights into the role of cooperativity in dynamic systems. How might these findings inform the design of more resilient polymer systems? Could the mechanisms demonstrated in the simulations be experimentally reproduced in synthetic systems? Expanding on these points would help bridge the gap between theoretical exploration and practical application, broadening the study's significance.

Minor comments:

The definition of cooperation should be defined more clearly. In what sense do molecule "cooperate" better when the dipole is introduced.

Are there any qualitative differences between Fig. 1a and 1b?

The use of "fascinating" is subjective and may perhaps be avoided altogether.

There are a number of typos, especially near the end of the manuscript.

Reviewer #2

(Remarks to the Author)

General Comments

The article models the response of self-assembling, supramolecular polymer systems to chemical stimuli, which is the introduction of chain-stopper monomers, using a minimal coarse-grained molecular dynamics model. The study compared two prototypical systems, where monomers can self-assemble cooperatively, denoted as Mcoop, or in a non-cooperative fashion, denoted as M. At equilibrium, M systems exhibit a unimodal size distribution around the most favorable aggregate size. In Mcoop, the cooperativity enhances the stability of large aggregates and leads to a bimodal size distribution with a significant proportion of free monomers. Upon stimuli, the aggregates in M respond homogeneously and reach a new unimodal size distribution. In Mcoop, the stimuli effects are largely absorbed by small aggregates, and large aggregates remain relatively stable and resilient to stimuli.

The simulation and analysis methodology is solid. Overall, this study provides a detailed characterization of different modes of response to stimuli and explains the microscopic origin of qualitatively different responses from two model systems. Although the conclusions are somewhat intuitive, they are consistent, well-explained, and well-supported by comprehensive evidence. Nevertheless, despite the consistency of the reasoning and conclusion presented above, the authors overlook several considerations, including some methodology details and perspectives from statistical mechanics.

Major concerns

1. Finite-size effect. One of the main arguments of this article is that the stimuli response must be considered as an emergent property of the entire polymer ensemble. This suggests that the observations coming from a finite simulation box might suffer from finite-size effects or be size-dependent. The authors need to address how varying box dimensions can influence their results.

2. Dynamics of rigid dipole rotor. The cooperativity is implemented by placing a freely rotating dipole in the center of the core bead of a monomer. The dipole is represented by a rigid rotor with two oppositely charged beads at its termini. It is implied that the charged beads are meant to modify the nature of intermolecular interactions, not to represent any actual part of the molecule/substance. For clarity and reproducibility, the authors are suggested to also specify the following information: What is the mass of these charge beads? Are these charged beads integrated using the same equation of motion as other beads? Do they feel any solvent friction as other beads do? This information is critical because mass and the equation of motion determine dynamics. How fast the dipoles adjust in response to their surroundings can have a profound influence. Ideally, to present a comprehensive investigation of the dynamics of the system, the authors should also quantify and compare the dipole reorientation response time, the characteristic rotational correlation times of a molecule both as a free molecule and within an aggregate, and the characteristic diffusion time of a monomer molecule.

3. Long-range electrostatics. Because the dipoles are explicitly represented by two connected charges, the dipole-dipole interaction is essentially calculated by the pairwise summation the Coulombic interaction, which is long-range in nature. However, in this study, all non-bonded interactions are truncated at a cutoff distance of 1.1 nm with no additional solver to account for the effect of long-range electrostatics. The authors should justify the decision to leave out long-range electrostatics.

4. The potential of mean force for dimerization. It is mentioned on page 7 that the rotating dipoles introduce an entropic penalty to dimerization because two diffusing free monomers must meet with favorable dipole orientation. While it seems plausible based on the equilibrium size distribution, this claim is not supported by quantitative evidence. The authors can compare the potential of mean force between two non-cooperative monomers with that between two cooperative monomers and see if that is the case. The authors are also suggested to validate if this claim is true or if this behavior persists when long-range electrostatics is added to the interaction energy.

5. Thermodynamics vs. Dynamics. I am concerned with the emphasis on dynamics in the work. The term “dynamics” is rather loosely defined here and seems to refer to the dynamic transition among states at an equilibrium, which deviates from what dynamics stand for to most people. The former mainly considers statistical probabilities, whereas the latter often involves time series analysis over a continuous trajectory. The work emphasizes how the observations emerge the “dynamics” of the systems but overlooks the fundamental and critical role of thermodynamics. Firstly, most evidence supporting the key findings are thermodynamic properties. Instead of showing time series analysis, the results are mostly based on the equilibrium size distribution and equilibrium transition probability matrix at equilibrium, which are derived from the underlying free energy landscape. Secondly, in the cooperative system, the authors observed resistance of large aggregates due to their stability and the absorbance of stimuli by small aggregates due to their abundance. While a detailed analysis of molecular traffic is vivid, the results are intuitive and can be readily deduced from equilibrium thermodynamics, or specifically the free energy landscape implied behind the probabilities. Finally, the article attributes the qualitatively different stimuli response to the difference in ‘dynamics’ between the M and Mcoop systems, but in fact the introduction of cooperative interactions alters rather the thermodynamics. If authors can investigate and compare the three characteristic/correlation timescales mentioned in Question 2, they can strengthen their argument about dynamics.

6. Terminology. The metaphorical concepts of “additivity” and “resource” might appeal to a broader audience, but they omit the important thermodynamic concept of free energy. For a scientific article, relevant scientific quantities should be addressed when applicable. For example, instead of using “additivity” or “resource” to reason the driving forces behind assembly and stimuli response behaviors, the authors can calculate the free energy change, perhaps ending up being highly nonlinear, for every additional monomer as an aggregate grows. The authors can further decompose the free energy into enthalpic and entropic contributions to provide an in-depth explanation. These quantitative physical quantities can further strengthen the discussion.

Minor remarks

1. Plot color and label. In many figures, the post-stimuli systems are labeled by the stimuli concentration, i.e., 40C, 120C, ..., 400C, whereas the pre-stimuli (or no stimuli) systems are labeled as M and Mcoop for the two respective systems. The authors can make the label more consistent for better clarity.

Version 1:

Reviewer comments:

Reviewer #1

(Remarks to the Author)

The authors have adequately addressed my previous comments, and the revisions have improved the clarity and completeness of the manuscript. I recommend the paper for publication.

Reviewer #2

(Remarks to the Author)

General Comments

I appreciate the authors' efforts in revising the manuscript and considering my comments. In the revised manuscript and supporting materials, the authors carefully address all the comments.

Comment 1: For finite size effect, the authors included additional sets of simulation at a smaller system size, demonstrating the emergence of the same feature. These results render the original simulation results, performed at a larger size, more robust.

Comment 2,3: The authors have supplied additional details regarding the implementation of rotating dipole and justified their methods on electrostatics interaction. The cooperativity was extensively characterized by calculating the diffusivities and rotational correlation times of monomers.

Comment 4: The authors have included additional calculation on the potential of mean force for dimerization, which quantitatively supports that the rotating dipoles introduce an entropic penalty to dimerization. The added calculation provided mechanistic insight into the cooperativity.

Comment 5,6: The authors have address comments regarding terminologies, which clarify the concepts to readers while maintaining scientific rigor.

Reviewer #1

Comments:

This manuscript describes the study of a dynamic network models where small molecules are able to assemble into large fibers due to attractive physical interactions. When introducing a small fluctuating dipole to the assembling agents, the dynamics are altered, promoting the aggregation in a similar structure, but with a bimodal distribution comprising a large number of highly dynamic units that dynamically interact with the rest of the network. While the emerging behavior of the cooperative and non-cooperative systems are quite similar, their response to external stimuli (here triggered by introducing capping agents) is very different. While this stimulus uniformly triggers a reduction of filament size in the non-cooperative system, it weakly affects the cooperative system. The reason is that the large number of small dynamic unit can shield the large filament against the action of capping agents. The authors therefore argue that the idea of cooperation is important for system resilience against changes in external environment. Generally, the paper is well-written, and results are presented in a convincing manner. The scientific approach is sound and nicely conducted.

Authors' response:

We thank the Reviewer for the positive comments on our work. We are glad that the Referee found our paper well written and interesting.

The comments below are mostly about the relevance of this work to either polymeric systems, or general complex systems.

1. Does the network under study represent any real physical system? Has the type of cooperativity observed in the simulations been reported before, either in natural or synthetic systems? Or is this primarily a toy model aimed at exploring the concept of cooperativity? Providing more context about the system's relevance to real-world scenarios would strengthen the impact of the study.

Authors' response:

We thank the Reviewer for this pertinent question. The minimalistic models used herein are certainly used to study collective dynamical behavior. However, despite their minimalistic nature, at the same time they also indeed reproduce key global features of real supramolecular polymer systems. As it has been discussed in detail in our previous work (Crippa et al. 2022, [10.1038/s41467-022-29804-5](https://doi.org/10.1038/s41467-022-29804-5)), despite their approximated character, such relatively simple models can render crucial aspects of the collective behavior of real supramolecular fiber systems, such as, e.g., BTA (Catekin et al. 2012, [10.1039/C2CS35156K](https://doi.org/10.1039/C2CS35156K); De Marco et al. 2021, [10.1021/acsnano.1c01398](https://doi.org/10.1021/acsnano.1c01398)), BTT (Demenev et al. 2010, [10.1021/cm902453z](https://doi.org/10.1021/cm902453z); Casellas et al. 2018, [10.1039/C8CC01259H](https://doi.org/10.1039/C8CC01259H)) or porphyrin-based supramolecular polymers (Fukui et al. 2017, [10.1038/nchem.2684](https://doi.org/10.1038/nchem.2684); Jung et al. 2018, [10.1021/jacs.8b06016](https://doi.org/10.1021/jacs.8b06016); Weyandt et al. 2022, [10.1038/s41467-021-27831-2](https://doi.org/10.1038/s41467-021-27831-2)). The introduction of chain-stopping molecules also represents a case of realistic supramolecular interaction, studied in several experiments, such as e.g. Sijbesma et al. 1997 ([10.1126/science.278.5343.1601](https://doi.org/10.1126/science.278.5343.1601)), Vantomme et al. 2019 ([10.1021/jacs.9b09443](https://doi.org/10.1021/jacs.9b09443)) and Weyandt et al. 2022 ([10.1038/s41467-021-27831-2](https://doi.org/10.1038/s41467-021-27831-2)). In particular, the use of rotating rigid dipoles, and of local dipole-dipole interactions, to reproduce the directionality and cooperative interaction between self-assembling monomers (such as BTA) in supramolecular fibers has been used, tested, and validated in previous simulation studies (see e.g. Bochicchio & Pavan, 2017, [10.1021/acsnano.6b07628](https://doi.org/10.1021/acsnano.6b07628); Bochicchio et al. 2017, [10.1038/s41467-017-00189-0](https://doi.org/10.1038/s41467-017-00189-0); De Marco et al. 2021, [10.1021/acsnano.1c01398](https://doi.org/10.1021/acsnano.1c01398)). Similarly,

the minimalistic representation employed in the models that we use herein aims at keeping the essential aspects of supramolecular polymer systems (as described in detail in: Crippa et al. 2022, [10.1038/s41467-022-29804-5](https://doi.org/10.1038/s41467-022-29804-5)), such as, e.g., the establishment of a dynamic supramolecular equilibrium via more (M_{coop} model) or less (M model) cooperative interactions and self-assembly mechanisms. Thanks to their essential coarse-grained character, the minimalistic models used herein allow extending the time and space-scales accessible via the MD integration, enabling a sound statistical characterization of the dynamical equilibrium state, which can be sampled well thanks to the large statistics of transition events made accessible by the description. In this sense

The simplicity of the monomers' structures and of the monomer-monomer interactions in the models used herein – which, despite offering lower detail in terms chemical description, at the same time capture the essential features of such self-assembling systems – makes it possible to reconstruct and study, at a qualitative level, collective as much as microscopic-level dynamical behaviors that would not be possible to study otherwise.

To better underline these points, in this revised version of our paper we introduced the following clarifying sentence in the main text at Pag. 5, line 14:

“These M and M_{coop} minimalistic models feature stochastic dynamics and simple monomer-monomer interactions that, despite the approximations, allow to qualitatively represent the essential features in terms of structural and dynamical behaviors of real supramolecular polymer systems (such as, e.g., BTA, BTT, porphyrin-based polymers in good solvents).[34,38,40] While the simplicity in the physical description provided by these models does not preserve a direct correspondence to specific chemical systems, it is useful to extend the time- and space-scales accessible by the MD simulations, and to attain general-level insights into collective dynamical behaviors of such complex self-organizing systems.”

2. While the authors convincingly argue that cooperativity enhances system resilience (as defined in their framework), it is not immediately clear how broadly this concept applies. Could there be cases where cooperativity might reduce resilience? Discussing potential counterexamples or boundary conditions for this concept would provide a more comprehensive understanding of the phenomenon.

Authors' response:

We thank the Reviewer for this comment. As explained in the answer to the previous point, the minimalistic models that we use herein are tuned to qualitatively reproduce the key features of dynamical supramolecular polymers in good solvents. While these models, as said, do not strictly correspond to any specific type of supramolecular polymer, the behaviors that we observe are consistent with those seen in realistic supramolecular polymer systems such as, e.g., BTA, BTT, or porphyrin-based supramolecular polymers, for example (this has been discussed in detail also in Crippa et al. 2022, [10.1038/s41467-022-29804-5](https://doi.org/10.1038/s41467-022-29804-5)).

The description of the monomers in these models is so abstract that, we believe, the obtained results are valid for supramolecular systems in general. Essentially, when we augment the cooperativity from model M to model M_{coop} , we show how in general the typical features of cooperative self-assembling systems emerge and become more prominent in the system, - e.g., a bimodal distribution of sizes, etc. Given the general character of these models, this indicates that the higher is the cooperativity, the more bimodal (non-uniform) will be the size distribution of the assemblies that are formed (i.e., the more the system will be populated by stable long assemblies and disassembled monomers) and, based on our results, the higher will be the global resilience of the system. In fact, our results show how, in these systems, the stimulus (cappers

binding) is mainly absorbed by the monomers (and by the smaller assemblies), and thus affects the “weaker” (smaller) entities to a greater extent as compared to the bigger and more stable fibers.

This question by the Reviewer is interesting. While here we make some stimulating comparisons/analogies between the minimalistic models used herein with existent supramolecular polymers types such as, e.g., BTA, BTT, porphyrin-based supramolecular polymers in good solvents, obviously, simulating all possible supramolecular polymer variants – or in general any different type of self-assembling system – is not feasible/sensible, and clearly exceeds the purpose of this specific work. However, we can comment that, based on the evidence obtained with such a minimalistic description of supramolecular polymers, the conclusions that we draw in this study appear to have a general character, and to be applicable to any type of supramolecular polymers system (e.g., in different context of solvents, temperature, chemical designs, etc.). Furthermore, given the abstract nature of the model and the generality of the obtained evidence, this concept seems thus applicable to virtually any type of self-assembling system where cooperativity might be present. The presence of free monomers in the equilibrium ensemble in fact is a thermodynamical consequence of the cooperativity in the polymerizing interactions. Provided that such a cooperativity is present (and that this does not tend to the infinite, so that basically the system would converge in forming just one single assembly – but this is an ideal case, unfeasible in practice), any self-assembling system would create an ensemble of assembled objects (stable) surrounded by weaker/smaller units that are highly unstable (sources of monomers), which would be then perturbed by an external stimulus like the addition of chain-stoppers by a greater extent than the more stable units, which would become in turn even more predominant than the smaller one under perturbation.

To better stress this relevant point, in this revised version of our work we have modified the comment below (at Page 24, line 9 in the original manuscript):

“In general, the results indicate how in a cooperative system, when the system undergoes stress, the stronger, larger entities become stronger as compared to the weaker smaller assemblies, while in a non-cooperative system the internal response to the perturbation is more homogeneous. Noteworthy, such different microscopic-level ways of responding to an external perturbation simply reflect the best (thermodynamic) ways for the two systems to evolve upon an external perturbation and to optimize the resources (the interactions).”

With the new text (at Page 25, line 20 in the revised manuscript):

“The minimalistic representation provided by the models used herein allows us to observe how the dynamical complexity of these supramolecular polymers’ systems, and the responsive behaviors that emerge from them are tightly dependent of the cooperativity in the monomer-monomer interactions. In general, the results indicate how in a cooperative system, when the system undergoes stress, the larger, more stable entities become stronger as compared to the weaker smaller assemblies, while in a non-cooperative system the internal response to the perturbation is more homogeneous. Noteworthy, such different microscopic-level ways of responding to an external perturbation simply reflect the best (thermodynamic) ways for the two systems to evolve upon an external perturbation and to optimize the resources (the interactions).

Qualitatively, this provides a relevant insight for supramolecular polymer systems in general: the higher is the cooperativity in the self-assembly, the more bimodal will be the distribution

of the assemblies (in terms of size and relative stability),[47-49] and the higher will be the resilience of larger entities as compared to smaller ones.”

3. The study offers interesting insights into the role of cooperativity in dynamic systems. How might these findings inform the design of more resilient polymer systems? Could the mechanisms demonstrated in the simulations be experimentally reproduced in synthetic systems? Expanding on these points would help bridge the gap between theoretical exploration and practical application, broadening the study's significance.

Authors' response:

Referring to the systems indicated in our response to point 1 of the Reviewer's comments, that is 1D supramolecular fibers formed by, e.g. BTA, BTT or porphyrin-based monomers, we can positively conclude that the cooperativity of monomers' self-assembly has a direct impact in promoting the resilience at the ensemble level to the perturbation induced by chain-stoppers binding. This represents a milestone for the rational design of such kind of supramolecular polymer systems, providing precious guidelines for e.g. the length control of supramolecular polymers via chain-stopper species (see e.g. Weyandt et al. 2021, [10.1038/s41467-021-27831-2](https://doi.org/10.1038/s41467-021-27831-2)).

More in general, in an example environment where the stimulus/perturbation is provided by the binding with competitive entities (like the cappers simulated herein), our results indicate that being able to rationally-design, and somehow encode, cooperativity into the self-assembly and in the monomer-monomer interactions would allow, in principle, to control the ensemble resilience to perturbation of supramolecular polymer systems. In this light, the rational design of such emergent property can be reconducted, in good approximation, to that of rationally design monomers that self-assemble cooperatively. The results that we report herein, together with those described previously, e.g., in Crippa et al. 2022, [10.1038/s41467-022-29804-5](https://doi.org/10.1038/s41467-022-29804-5), and all our previous results acquired in the study of supramolecular polymers across the years, indicate that to have cooperativity, it is fundamental to have some sort of competition. For example, monomers that preferentially assume a non-assembling conformation when they are disassembled in solution, but that can undergo transition to an assembling state at an affordable thermodynamical cost, are good candidates to generate cooperative self-assembling systems. This is the case, e.g., of BTA motifs, where the amide bonds are in plane (the most favorable configuration for amides bound to the aromatic core) in the disassembled state, but in a fiber are tilted and engaged in three-fold hydrogen bonding helices. Below a certain “critical nucleus/size”, the amides are more favored to stay planar while, above a certain size, the stabilization provided by the H-bonding network is strong enough to overcome the tendency to return planar (and to break the monomer-monomer interactions). Similar effects are seen in monomers that assume preferentially a folded and non-polymerizable conformation when disassembled in solution, and an extended planar (disc-like) conformation, prone to directional polymerization via stacking, which is slightly higher in free-energy (unfavorable). All these cases suggest viable molecular-level approaches to control cooperativity in the self-assembly, offering opportunities to control, in turn, the features of the ensembles (distribution of sizes, their dynamicity, etc.), and the collective resilience properties that may emerge within them. While closing this circle is not easy, we believe that this is possible, and this specific work constitutes an important piece of the pathway to reach this ambitious goal.

We agree with the Reviewer that this is a highly relevant point, worth of being mentioned. To this end, in this revised version of our paper, we have added the following sentence in the Conclusions section (at Page 26, line 19):

“Specifically, the results obtained by our models indicate how the design of monomers that can interact and self-assemble cooperatively in a controllable way may offer the opportunity to control collective, emergent properties, for example determining the resilience to the perturbation induced by chain-stoppers.”

Minor comments:

The definition of cooperation should be defined more clearly. In what sense do molecule “cooperate” better when the dipole is introduced.

Authors’ response:

In this revised version of our work, we have extended the explanation and discussion on the dipole effect on the cooperativity of monomer-monomer interactions and self-assembly, adding further qualitative and quantitative insights that better clarify the origin of cooperativity in the models we use herein. We have added two new Figures S11 and S12 in which we report evidence of cooperativity. In general, these additional analyses demonstrate how the presence of a freely rotating dipole introduces an entropic penalty that has to be overcome to trigger self-assembly in the \mathbf{M}_{coop} system. In detail, such rotating dipole is free to rotate around the center of the monomer, so that the probability that 2 monomers stack having the two dipoles oriented perpendicularly to their plane and complementary to each other (+ vs. – exposed charges) is relatively low. This results in an entropic penalty to self-assembly, introducing a critical nucleus to fiber elongation in the system (typical of cooperative systems). In a longer oligomer where the dipoles of the monomers are all aligned and interacting with each other, the monomers at the fiber ends are in a more favorable configuration for the binding of an additional monomer with respect to free monomers with random dipole orientation. This is demonstrated by additional data reported in the two new Figures S11 and S12, mentioned in this revised version of our paper at Page 7, lines 15 in the following sentences:

This introduces an entropic penalty to dimerization, due to the relatively low likelihood that two free monomers stack with the dipoles oriented to favor binding. This is shown in Figures S12a-c, which report the free-energy of dimer formation in the \mathbf{M} and \mathbf{M}_{coop} systems, evaluated via Well-tempered Metadynamics[37,40,41], and its decomposition into enthalpic and entropic terms (see Methods section for details). The polymerization of longer oligomers/fibers, in which the dipoles at the ends are already aligned is thus favored compared to that of monomers/smallest assemblies, where dipoles have random or weak alignment and their interactions are inherently less stable and more dynamic (see Figure S12d). This confers to \mathbf{M}_{coop} the typical features of a cooperative supramolecular system, such as, e.g., the presence of a critical nucleus for fiber elongation, and a characteristic bimodal size distribution (see below).“

and at Page 22, line 7:

“The difference in the ensemble behavior of cooperative vs. non-cooperative systems lies in the concept of super-additivity. In a cooperative system, interactions are super-additive, promoting the growth of larger aggregates and unfavoring aggregates smaller than a critical size (see also the free-energy profiles in Figures S11-12, associated to the polymerization equilibrium of the two systems).”

Concerning cooperativity see also response to Reviewer 2.

Are there any qualitative differences between Fig. 1a and 1b?

Authors' response:

The two figures represent the monomer-monomer interactions of **M** and **M_{coop}** monomers. In the **M** model, the monomer-monomer interaction is fully driven by the core-core LJ potential (black curve). In the **M_{coop}** model, the monomer-monomer interaction results from the combination of a core-core LJ (red) and a directional electrostatic interaction driven by the alignment of the two rotating dipoles present monomers' cores (yellow). Two snapshots of the simulated systems at the equilibrium are reported, intuitively showing similar aggregated fibers (the differences are then extensively quantified in the following of the manuscript).

The use of "fascinating" is subjective and may perhaps be avoided altogether.

Authors' response:

We thank the Reviewer for this comment. We removed the term, and modified the sentences in a more objective fashion.

There are a number of typos, especially near the end of the manuscript.

Authors' response:

We thank the Reviewer for pointing this out. In this revised version of our paper, we have carefully reviewed the text to correct the typos present in the initial submission.

Reviewer #2

General Comments:

The article models the response of self-assembling, supramolecular polymer systems to chemical stimuli, which is the introduction of chain-stopper monomers, using a minimal coarse-grained molecular dynamics model. The study compared two prototypical systems, where monomers can self-assemble cooperatively, denoted as **M_{coop}**, or in a non-cooperative fashion, denoted as **M**. At equilibrium, **M** systems exhibit a unimodal size distribution around the most favorable aggregate size. In **M_{coop}**, the cooperativity enhances the stability of large aggregates and leads to a bimodal size distribution with a significant proportion of free monomers. Upon stimuli, the aggregates in **M** respond homogeneously and reach a new unimodal size distribution. In **M_{coop}**, the stimuli effects are largely absorbed by small aggregates, and large aggregates remain relatively stable and resilient to stimuli. The simulation and analysis methodology is solid. Overall, this study provides a detailed characterization of different modes of response to stimuli and explains the microscopic origin of qualitatively different responses from two model systems. Although the conclusions are somewhat intuitive, they are consistent, well-explained, and well-supported by comprehensive evidence.

Authors' response:

We thank the Reviewer for underlining the value of the research presented in the manuscript.

Nevertheless, despite the consistency of the reasoning and conclusion presented above, the authors overlook several considerations, including some methodology details and perspectives from statistical mechanics.

Major concerns

1. Finite-size effect. One of the main arguments of this article is that the

stimuli response must be considered as an emergent property of the entire polymer ensemble. This suggests that the observations coming from a finite simulation box might suffer from finite-size effects or be size-dependent. The authors need to address how varying box dimensions can influence their results.

Authors' response:

We thank the Reviewer for this pertinent comment. Indeed, we agree with the Reviewer that the system's size might play an important role in such types of simulations. Nonetheless, the results and conclusions drawn from our study are not affected by finite size issues. To prove this, in this revised version of our work (in the SI), we report analogous simulations and analyses conducted, for comparison, on smaller model systems containing $N=500$ monomers each. The monomer concentration in these smaller models is the same concentration as in the larger $N=2000$ monomers systems (that we report in the main paper, and which offer a higher statistical robustness in the extracted results). We also underline that, despite the minimalistic description of the monomers in these models, the $N=2000$ monomers case already requires a considerable computational effort in order to reach sufficiently long simulation times and sufficiently robust sampling to quantify with satisfactory statistics the quantities monitored herein (so that testing also larger systems is unpractical). Anyways, this comparison between the $N=500$ and $N=2000$ monomers systems is already very useful to prove the robustness of the obtained results and of the general character of the conclusions.

Despite lower statistics for the $N=500$ monomers systems, globally, the behavior emerging from the comparison of the cooperative (\mathbf{M}_{coop}) and non-cooperative (\mathbf{M}) model is consistent with the results obtained with $N=2000$ models. In particular, as it can be now seen in new Figs. S7-S9, also in the $N=500$ monomers case, the \mathbf{M}_{coop} vs. \mathbf{M} model systems present the same differences in terms of global features, such as, e.g., in terms of unimodal vs. bi-modal, of the absence vs. presence of a critical nucleus for fibers' growth, of uniform vs. non-uniform impairment of the polymerization/depolymerization equilibria across the sizes that populate the systems, etc. In general, independently of the number of monomers and on the size of the systems, we do observe the same distinctive features in the \mathbf{M}_{coop} vs. \mathbf{M} systems and the same global differences between them. This proves that the results obtained on the $N=2000$ monomers systems are not impacted significantly by finite-size effects.

We believe that including such smaller size systems for comparison strengthens our general conclusions, and makes the observation based on the data extracted from the larger $N=2000$ systems that we show in the main paper even more robust.

In this revised version of our paper we added the following sentence at Page 10, line 20:

"We also performed the same comparison between smaller \mathbf{M} and \mathbf{M}_{coop} systems containing 500 monomers, at the same concentration (1/4 cell volume). Such smaller size systems report the same distinctive features for the cooperative (\mathbf{M}_{coop}) and non-cooperative (\mathbf{M}) systems, proving that the observed behaviors are not significantly impacted by finite-size effects (see Figure S7)."

And at Page 22, line 25:

"Moreover, analogous behavior emerges in smaller-scale systems containing 500 monomers when perturbed by the chain-stopper stimulus, confirming that our results are not vitiated by finite-size effects (see Figures S8-S9) and, globally, the robustness and generality of our conclusions."

Details on the N=500 monomer systems have been also added in the methods section of the revised manuscript at Page 29, line 13:

*“An additional set of simulation of a smaller model system, containing N=500 monomers (in a cubic simulation box of volume 22.056x22.056x22.056 nm³), having the same concentration of monomers and of the larger (N=2000) systems, has been carried out. The dynamical equilibrium of such smaller **M** and **M_{coop}** variants has been analyzed as done for the larger systems. When perturbing these smaller model systems, the same copper concentrations as in the larger analogs was maintained (adding 10, 30, 50, 75 and 100 Cs). These models demonstrated consistent behaviors as those seen in the larger N=2000 systems (see Figures S7-S9).”*

2. Dynamics of rigid dipole rotor. The cooperativity is implemented by placing a freely rotating dipole in the center of the core bead of a monomer. The dipole is represented by a rigid rotor with two oppositely charged beads at its termini. It is implied that the charged beads are meant to modify the nature of intermolecular interactions, not to represent any actual part of the molecule/substance. For clarity and reproducibility, the authors are suggested to also specify the following information: What is the mass of these charge beads? Are these charged beads integrated using the same equation of motion as other beads? Do they feel any solvent friction as other beads do? This information is critical because mass and the equation of motion determine dynamics. How fast the dipoles adjust in response to their surroundings can have a profound influence. Ideally, to present a comprehensive investigation of the dynamics of the system, the authors should also quantify and compare the dipole reorientation response time, the characteristic rotational correlation times of a molecule both as a free molecule and within an aggregate, and the characteristic diffusion time of a monomer molecule.

Authors' response:

We thank the Reviewer for asking such clarifications. In this revised version of our paper, the missing details have been added in the methods section (moreover, the molecular models used for all simulations, with all their parameters, are available on GitHub, and will be on Zenodo after publication). The mass of the dipole beads is 1/3 the mass of that of the monomer beads (i.e., the addition of the dipole increases the monomer mass by ~10%), and their motion is integrated following the same equations of motions of all the other beads (same friction). The dipole beads do not bear VdW interactions, so that they can rotate around the center of the monomers without bumping into other beads. Their positions are anchored to the central bead of the monomers via constraints keeping them at 0.14 nm. Similar use of rotating dipoles to mimic directionality and cooperativity in supramolecular polymers models have been previously adopted in successful way (see e.g., Bochicchio et al., ACS Nano 2017, 11, 1000; Bochicchio et al. J. Phys. Chem. Lett. 2017, 8, 3813; Bochicchio et al. Nature Commun. 2017, 8, 147; De Marco et al. ACS Nano 2021, 15, 14229; Crippa et al. Nature Commun. 2022, 13, 2162).

In such models the rotation motion of the dipoles is faster than the motion of the monomers. When two disassembled monomer cores collide, but with their dipoles not oriented in such a way to stabilize the dimerization, the monomer-monomer interactions are relatively weak and have a low survival lifetime. Altogether, in the case this provides a **M_{coop}** model where two free monomers have a relatively low probability to stack with their dipoles oriented properly to

establish stable interactions (statistical/entropic penalty to oligomers nucleation), while at the same time this probability increases considerably when one disassembled monomer finds, e.g., the tip of a preexistent long fiber, where the tip monomer has a pre-oriented dipole already in orthogonal configuration, more favorable to fiber elongation.

As discussed in the Results section, the role of the central rotating dipoles is to provide directionality to the monomer-monomer interactions and to impart cooperativity to the polymerization. This is granted by an additional degree of freedom (the dipoles rotation) that allows an optimal interaction when fibers larger than the critical nucleus are formed (i.e., fibers in which the dipoles are stably oriented perpendicular to the monomer planes, prone to maximize the interaction and to fiber elongation), while below such critical size smaller assemblies are unstable and the probability for dipoles orientation is lower. The addition of such rotating dipoles aims at imparting cooperative features to the polymerization of \mathbf{M}_{coop} , that emerge only locally, when the monomers are in close contact to each other), and it does not aim to reproduce any real electrostatic interaction. For this reason, and in order to maintain the character of the dipole-dipole interactions local, no long-range electrostatics are included in the simulations.

While upon original submission all used monomers (and their parameters) were available in the github, we did not also enclose all the relevant details in the Methods section. Complete details of the parameters used for the \mathbf{M} and \mathbf{M}_{coop} beads monomers are provided in this revised version of the manuscript at page 27, line 16:

“The \mathbf{M}_{coop} model has the same hexagonal topology and bonded interactions of the \mathbf{M} model, with the addition of a dipole centered on the central monomer bead composed of two charged beads ($q = \pm 1.4 e$) constrained at fixed distance from each other of $d=0.28$ nm (each placed 0.14 nm from the center of the core bead). The two beads are kept aligned along a straight vector (see Figure 1b) via a harmonic angular potential that maintains an angle of 180° between the charged beads and the core bead (with force constant of 1500 kJ/mol). Such dipole charged beads have mass 1/3 of the mass of the other (regular) beads and interact only via Coulomb interactions (no van der Waals interactions). In this way, charged beads do not bear excluded volume, forming a dipole that is free to rotate around the center of the molecule (without any interaction with the other monomers beads) and that can interact only with the dipoles of the other monomers.

The addition of dipoles implies an entropic penalty for the binding of two free \mathbf{M}_{coop} monomers, having randomly oriented dipoles, as compared to the binding of \mathbf{M}_{coop} monomers (or fibers) to a pre-assembled fiber, where the dipole vectors at the ends are optimally oriented to favor attraction.

Therefore, as discussed in the Results section, rotating dipoles enhance monomer-monomer interaction directionality and offer additional degrees of freedom that favor the formation of fibers larger than a critical nucleus, thus imparting cooperativity. No long-range electrostatics are included in the simulations in order to preserve the local character of the dipole-dipole interactions and to avoid spurious long-range attractions/orientations in the \mathbf{M}_{coop} model.”

Furthermore, at page 29, line 22, we also added the following clarifications:

“To simulate dynamics, implicit-solvent Langevin equations are solved for all the beads, in which the stochastic term implements both the solvent friction and thermal fluctuations of the system.”

To reinforce such observations and also in compliance with the suggestions of the Reviewer, we also performed additional analyses.

We performed extra MD simulations of the \mathbf{M} and \mathbf{M}_{coop} models (containing 2000 monomers each), prolonging the trajectories by 5 extra μs (i.e. collecting extra sampling of the systems *at equilibrium*). We sampled the trajectory with higher frequency (every 50 ps instead of 300 ps) to have a better insight on the dynamics.

We also performed 5 μs of two new variants of the \mathbf{M} and \mathbf{M}_{coop} systems, where LJ is replaced by a Weeks Chandler Anderson potential (truncated and shifted at the LJ minimum) with the same parameters, and dipole charges were removed, so that only the excluded-volume interactions are maintained. We name these repulsive variants \mathbf{M}^{R} and $\mathbf{M}_{\text{coop}}^{\text{R}}$. These systems provide a reference of the monomer and dipole dynamics in a framework where no binding takes place and monomers remain free. Such repulsive variants have been then used to compare the dynamics of monomers diffusion and rotation, and the dipole rotational dynamics.

We computed diffusivities of the systems \mathbf{M} , \mathbf{M}_{coop} and \mathbf{M}^{R} , $\mathbf{M}_{\text{coop}}^{\text{R}}$ observing that the presence of dipoles slightly slows down diffusion. This is due to the higher monomer mass, as shown by the $\sim 8\%$ decrease in diffusivity between \mathbf{M}^{R} and $\mathbf{M}_{\text{coop}}^{\text{R}}$. When comparing \mathbf{M} and \mathbf{M}_{coop} diffusivities we observe a decrease of $\sim 25\%$, which depends also on the different size of the aggregates that form. While such variations in diffusivity do have any important effect in practice on the validity of the observations we report herein, these additional results have been now also reported in Fig. S10a-b for completeness.

We then evaluated the rotational correlation times of the monomers, and, where present, of the dipole vectors. This allowed us to assess the average rotational dynamics of the systems. The \mathbf{M}^{R} and $\mathbf{M}_{\text{coop}}^{\text{R}}$ monomers exhibit the same rotational dynamics, which is much slower than that of the free dipole vectors in $\mathbf{M}_{\text{coop}}^{\text{R}}$. The rotational autocorrelation functions in Fig. S10c-d show that the characteristic rotational time of the dipole is around 2 ps, whereas the rotational time of monomers is around 94 ps, demonstrating that the dynamics of the monomers is ~ 50 times slower than that of the dipoles.

These data suggest that, when the coulomb interactions are active, the orientational response of *free* dipoles is significantly faster than that of *free* monomers. Of course, the average rotational dynamics changes when we consider the interacting systems at the equilibrium, i.e. \mathbf{M} and \mathbf{M}_{coop} . The rotational autocorrelation of monomers and dipole vectors indicate that when monomers aggregate forming the equilibrium fiber distribution have a much slower rotational dynamics: the characteristic time of monomers in \mathbf{M} is of 210 ns, whereas in \mathbf{M}_{coop} the characteristic time of monomers is about 480 ns, together with a much slower rotation time of the associated dipole vectors (410 ns). The difference in rotational dynamics between \mathbf{M} and \mathbf{M}_{coop} is also consistent with the shorter persistence length of the former, already underlined by LENS analysis.

We added these additional results in the revised version of the SI (Fig. S10), and we also added at Page 10, line 26, the following clarification sentence:

“Furthermore, to enrich the comparison between the \mathbf{M} and \mathbf{M}_{coop} systems we provide further details on their diffusion and rotation dynamics at the equilibrium, and on the effect of the formation of the polymer network. The results, reported in the SI (Figure S10), show that the dipoles have a small impact on the dynamics of free monomers, but significant differences between \mathbf{M} and \mathbf{M}_{coop} dynamics emerge when the equilibrium fiber distribution is formed.”

We also properly defined the \mathbf{M}^{R} and $\mathbf{M}_{\text{coop}}^{\text{R}}$ variants in the revised Methods section (at Page 28, line 25):

“For supporting analyses, we also performed simulations of “purely repulsive” variants of \mathbf{M} and \mathbf{M}_{coop} models, that we name \mathbf{M}^R and \mathbf{M}_{coop}^R , respectively. These systems mirror the \mathbf{M} and \mathbf{M}_{coop} models, except for the fact that attractive interactions have been removed, by truncating and shifting the LJ potentials at the minimum ($r=2^{1/6} \sigma$) and by setting the dipole charges to 0. The simulation of \mathbf{M}^R and \mathbf{M}_{coop}^R is useful to compare the dynamics of free monomers as to that of the interacting systems at equilibrium (see Fig. S10).”

We also described the setup of these additional analyses in the new Methods section (Page 30, line 15 of the revised manuscript):

“Further CG-MD simulations were performed for the \mathbf{M} and \mathbf{M}_{coop} models, to assess the average diffusion and rotation dynamics of the monomers at equilibrium. Additional 5 μ s of MD were run starting from the final equilibrium configuration of the production runs. A faster sampling time of 50 ps was adopted here, to have a better detail on the dynamics. Similarly, 5 μ s of MD were performed for the “purely repulsive” \mathbf{M}^R and \mathbf{M}_{coop}^R , following an equilibration stage of 2500 ps (sufficient for non-attractive monomers). For the latter system we also performed 50 ns of MD with a 0.5 ps sampling time, to fully capture the fast rotation dynamics of the free dipoles.”

Finally, we have also indicated the methodology of these analyses in the new Methods section at Page 33, line 6:

“The diffusivity of monomers in the \mathbf{M} and \mathbf{M}_{coop} , and in the purely repulsive variants \mathbf{M}^R and \mathbf{M}_{coop}^R is obtained from the average mean-squared-displacement (MSD) of all monomers’ centers of mass (Fig. S10a-b) as a function of the displacement time τ . The diffusion constant D is then obtained from the fitted angular coefficient of the MSD via Einstein’s relation ($MSD(\tau)=6D\tau$).

Finally, rotational autocorrelation functions of the monomers are computed by associating to each monomer a vector perpendicular to its plane and evaluating the vector autocorrelation function in time. Same procedure was followed for the dipole vectors in the \mathbf{M}_{coop} (and \mathbf{M}_{coop}^R) systems.”

3. Long-range electrostatics. Because the dipoles are explicitly represented by two connected charges, the dipole-dipole interaction is essentially calculated by the pairwise summation the Coulombic interaction, which is long-range in nature. However, in this study, all non-bonded interactions are truncated at a cutoff distance of 1.1 nm with no additional solver to account for the effect of long-range electrostatics. The authors should justify the decision to leave out long-range electrostatics.

Authors’ response:

As clarified above, while long-range effects are key for a correct treatment of electrostatic phenomena, in this specific case these are not relevant. In fact, as clarified, in the minimalistic models used herein, the rotating dipole is just “a trick” to impart cooperativity to self-assembly and a critical size for nucleation dictated by the entropic penalty to monomer-monomer interaction stabilizations as a function of the aggregates size. For this reason, the interaction of the dipoles is kept local and long-range electrostatic corrections are not considered in this model in order to avoid spurious effects that are undesired in this model.

We clarified this point in the Methods section of this revised version of the manuscript (Page 28, line 7 of the revised manuscript), as reported above and also here below again for the best clarity:

“No long-range electrostatics are included in the simulations in order to preserve the local character of the dipole-dipole interactions and to avoid spurious long-range attractions/orientations in the M_{coop} model.”

4. The potential of mean force for dimerization. It is mentioned on page 7 that the rotating dipoles introduce an entropic penalty to dimerization because two diffusing free monomers must meet with favorable dipole orientation. While it seems plausible based on the equilibrium size distribution, this claim is not supported by quantitative evidence. The authors can compare the potential of mean force between two non-cooperative monomers with that between two cooperative monomers and see if that is the case. The authors are also suggested to validate if this claim is true or if this behavior persists when long-range electrostatics is added to the interaction energy.

Authors' response:

We thank the Reviewer for raising this relevant point. In compliance of this observation, in this revised version of our work, we have conducted additional analyses in this sense, which support this statement and, we believe, provide more robustness to our work.

In compliance to this comment, and to prove the cooperativity of the monomer-monomer interaction in the M_{coop} model, we evaluated the free energy of binding of two monomers, evaluating also the enthalpic and entropic components associated to it. To this purpose we (i) employed Well-tempered Metadynamics [Barducci, et al. *Phys Rev Lett*, **2008**, *100*, 020603] (WT-MetaD) simulations to obtain free energy profile as a function of the core-core distance between the monomers, (ii) we then decomposed the free-energy (ΔF) profile in terms of the internal energy (ΔU) and entropic ($-T\Delta S$) profiles following the same procedure described in detail in Refs. (Gimondi, et al. *J. Chem. Phys.* **2018**, *149*, 104104, Kollias, et al. *Adv. Theory Simul.* **2020**, *3*, 2000092, Leanza, et al. *Chem. Sci.* **2023**, *14*, 6716). The results show that indeed the difference between dimerization free energies in the two systems has an entropic origin.

In this revised version of our work, these new additional results are reported in the new Fig. S12a-c and the discussion about cooperativity has been modified, referring also to these new insights (Page 7 line 15 of the revised manuscript):

This introduces an entropic penalty to dimerization, due to the relatively low likelihood that two free monomers stack with the dipoles oriented to favor binding. This is shown in Figures S12a-c, which report the free-energy of dimer formation in the M and M_{coop} systems, evaluated via Well-tempered Metadynamics [37,40,41], and its decomposition into enthalpic and entropic terms (see Methods section for details). “

Regarding long-range electrostatics, as motivated in the response to previous point, the dipole-dipole interactions in this model do not aim at reproducing any real (or realistic) electrostatic interaction, but are just used as a (validated) proxy for introducing cooperativity in the model and the possibility of locally enhancing the directional interactions between the monomers, as done in precedence in various coarse-grained models of analogous supramolecular polymers systems (see e.g., Bochicchio et al., *ACS Nano* **2017**, *11*, 1000; Bochicchio et al. *J. Phys. Chem. Lett.* **2017**, *8*, 3813; Bochicchio et al. *Nature Commun.* **2017**, *8*, 147; De Marco et al. *ACS Nano*

2021, 15, 14229; Crippa et al. *Nature Commun.* **2022**, *13*, 2162). This mimics effects such as, to mention a few realistic examples, of monomers bearing hydrogen bonding groups that can optimize them locally when they come at short distance from each other, or monomers with flat/aromatic groups that are folded when disassembled in solution (no-directional interactions between them), but that can rearrange when they get in tight contact with each other enhancing, e.g., directional core-core interactions (this is the case, e.g., of BTA supramolecular polymers – see: Bochicchio et al., *ACS Nano* **2017**, *11*, 1000; Bochicchio et al. *J. Phys. Chem. Lett.* **2017**, *8*, 3813; Bochicchio et al. *Nature Commun.* **2017**, *8*, 147; De Marco et al. *ACS Nano* **2021**, *15*, 14229).

In the Methods of the revised version of the manuscript we have added a specific section relative to the Free-energy calculations performed via WT-MetaD (page 34 line 22 of the revised manuscript):

“Well-Tempered Metadynamics Calculations

The free-energy profiles associated to the dimerization in the \mathbf{M} and \mathbf{M}_{coop} systems have been evaluated via Well-Tempered Metadynamics (WTMetaD) [41] simulations, that enhance the sampling of the system conformations allowing the accurate estimation of ensemble averages in reduced simulation times. We simulated systems containing 2 \mathbf{M} (or \mathbf{M}_{coop}) monomers in a box of size 3.5012x3.5012x3.5012 nm³, having the same concentration of the unperturbed $N=2000$ monomer systems. The WT-MetaD history dependent bias potential was applied on the distance between the monomer centers, stimulating the binding and unbinding.[35-37,40] The biasing potential was built by depositing every 10 ps Gaussian kernels of width equal to 0.1 nm and initial height of 0.1 kJ/mol. A bias factor of 5 was chosen to regulate the decrease in the kernel height, and the convergence of the deposited bias. In 1 μ s of WT-MetaD, both systems efficiently sampled binding/unbinding events, converging to the dimerization free-energy profiles shown in Figure S12a. Following the protocols described in Refs. [55-57], we have then also decomposed the free-energy (ΔF) in its internal energy (ΔU) and entropic ($-T\Delta S$) terms (see Figures S12b-c). Finally, in the case of the \mathbf{M}_{coop} model, we also estimated the free-energy difference associated to fiber elongation by exploring with WT-MetaD the binding/unbinding between two monomers, one of which has its dipole vector restrained (via harmonic angular potentials) along the direction orthogonal to the monomer plane, as it would be the case for a monomer located at the tip of a fiber. The result, in Figure S12d, shows that the free-energy gain associated to fiber growth via monomer addition is larger than the gain determined by the association of two disassembled monomers, supporting the cooperativity of the self-assembly in the \mathbf{M}_{coop} model.”

5. Thermodynamics vs. Dynamics. I am concerned with the emphasis on dynamics in the work. The term “dynamics” is rather loosely defined here and seems to refer to the dynamic transition among states at an equilibrium, which deviates from what dynamics stand for to most people. The former mainly considers statistical probabilities, whereas the latter often involves time series analysis over a continuous trajectory. The work emphasizes how the observations emerge the “dynamics” of the systems but overlooks the fundamental and critical role of thermodynamics. Firstly, most evidence supporting the key findings are thermodynamic properties. Instead of showing time series analysis, the results are mostly based on the equilibrium size distribution and equilibrium transition probability matrix at equilibrium, which are derived from the underlying free energy landscape. Secondly, in the cooperative system, the authors observed resistance of large aggregates due to their stability and the absorbance of stimuli by small aggregates due to their abundance. While a detailed analysis of

molecular traffic is vivid, the results are intuitive and can be readily deduced from equilibrium thermodynamics, or specifically the free energy landscape implied behind the probabilities. Finally, the article attributes the qualitatively different stimuli response to the difference in ‘dynamics’ between the M and M_{coop} systems, but in fact the introduction of cooperative interactions alters rather the thermodynamics. If authors can investigate and compare the three characteristic/correlation timescales mentioned in Question 2, they can strengthen their argument about dynamics.

Authors’ response:

We thank the Reviewer for stimulating the discussion on this point. As already mentioned in the answer to the 2nd question, our message here is mostly focused on the *polymerization/depolymerization* dynamics, that is the binding-unbinding transition dynamics that characterizes the molecular traffic across the fiber network *at the equilibrium*. In this sense we indeed compare equilibrium states, under or free from perturbation, but we investigate how the local, microscopic-level dynamics that animates such supramolecular equilibrium is affected by the perturbation. We underline that this is relevant, and that ensemble averaged quantities may be insufficient to understand the origin of emergent stimuli-responsive behaviors, as recently demonstrated e.g. in supramolecular colloidal superlattices, where the emergence of collective properties can be explained by a microscopic-level description of the internal dynamics of the system (see, e.g., Lionello et al. ACS Nano 2023, 17, 275-287). In other words, we are not following the dynamic drift of a whole system brought out-of-equilibrium, but how the *internal dynamics*, in terms of molecular exchange between its components, changes from the unperturbed to the perturbed equilibrium states – i.e., the subject is not the ensemble as a whole, but the internal dynamical network that characterizes it.

The analyses reported in the manuscript provide extensive analyses of the internal dynamical complexity of such self-assembling system. All the transition matrices, Sankey diagrams and the LENS descriptor time-series analyses (see e.g. Crippa et al., PNAS, 2023, 120 e2300565120) indeed elaborate dynamic information, quantifying how monomer polymerization/depolymerization states/transitions communicate with each other at every successive time interval, for all time intervals sampled along the MD trajectories.

While it is true that the different resilience of the **M** and **M**_{coop} systems manifests in reaching different new equilibrium structure of the two systems under perturbation, here we also show how the *local microscopic dynamics* that govern the polymerization-depolymerization propensities of the assemblies that populate the systems (in a microscopic level rather than macroscopic analysis) change as the perturbation entity increases in a cooperative vs in a non-cooperative system.

As already introduced in our previous work (Crippa, et al. Nat. Commun 2022, 16, 2162), understanding the microscopic-level molecular exchange dynamics of these systems, and how it relates to the monomer structure and interactions on the one hand, and to the global, thermodynamic properties of the system on the other hand, is essential toward understanding how to rationally control and design collective stimuli-responsive properties. In fact, our results show how such microscopic level analyses allow to monitor how such kind of collective behaviors may arise and be amplified under perturbation. This cannot be done with macroscopic/ensemble approaches (where one would see the system responding, but with no sufficient resolution to understand where such stimuli-responsiveness originate from, nor the mechanisms controlling it). In this specific case, for example, our conclusions on cooperativity and resilience offer new insight toward the rational design of complex dynamical systems with

an internal dynamical networks that can resist or react/respond to external perturbation in controlled way.

In the revised version of the manuscript we adjusted two sentences to better clarify this fundamental point when it is discussed in detail.

First , at page 19 line 3 of the revised manuscript:

“Stimuli-responsiveness emerges from the internal, microscopic dynamics underpinning the systems, i.e. from the “local features” of the network of molecular exchange between all the different-size aggregates at the equilibrium

At the same time, this “resilience” of the fibers is not simply a property of the individual monomers or assemblies, but rather a collective property emerging from the network of communications that connects all the entities that populate the system.

For this reason, understanding the origin of such collective stimuli-responsive behaviors, and of the mechanisms underpinning their amplification, requires a microscopic-level analysis of these complex systems (as standard ensemble averages alone do not capture where such behaviors originate from, nor the processes controlling them).”

And again at page 19 line 24 of the revised manuscript:

“We can quantify how the intensity of the stimulus (the C concentration) affects the internal dynamics of exchange between aggregates, by evaluating how the probability of polymerization/depolymerization of all the different-size entities that populate the systems change under perturbation with respect to the unperturbed equilibrium conditions.”

6. Terminology. The metaphorical concepts of “additivity” and “resource” might appeal to a broader audience, but they omit the important thermodynamic concept of free energy. For a scientific article, relevant scientific quantities should be addressed when applicable. For example, instead of using “additivity” or “resource” to reason the driving forces behind assembly and stimuli response behaviors, the authors can calculate the free energy change, perhaps ending up being highly nonlinear, for every additional monomer as an aggregate grows. The authors can further decompose the free energy into enthalpic and entropic contributions to provide an in-depth explanation. These quantitative physical quantities can further strengthen the discussion.

Authors’ response:

We thank the Reviewer for these observations that, we believe, allowed us to improve our work and to make it more robust.

While terms like additivity and cooperativity are widely adopted in the context of multivalent interactions and self-assembling systems, we fundamentally agree that, for what concerns the thermodynamics of the two systems, providing more direct insights in terms of free-energy of polymerization could be useful. To this end, and also to prove the cooperativity in the system and the entropic penalty associated to assembly nucleation in the M_{coop} system we performed additional analyses in this sense, which are reported in this revised version of our work in new Figs. S11 and S12.

The landscapes of polymerization-vs.-depolymerization in the systems' components provide a clear quantitative insight into the behaviors and thermodynamics of these systems.

In the case of our simulations, such insights can be directly attained from the assembly size distributions of Fig. 1c, for example. Given that the equilibrium is reached and well sampled in these simulations, these population distributions contain information on the relative probabilities for observing different-size entities in the systems in equilibrium conditions.

We can compute the probability of a monomer to be in an aggregate of certain size (P_{size}) and use it to estimate a free-energy landscape associated to the population of different aggregate states at equilibrium (as: $F_{\text{size}} = -kT \ln(P_{\text{size}})$).

In this revised version of our work, in the new Figs. S11a-b, we report both the monomer distributions (in % over 2000 monomers in total) among different aggregated states (Fig. S11a) and their relative free-energy estimate (Fig. S11b).

The \mathbf{M}_{coop} system shows a local free-energy minimum at the monomeric state, followed by a free-energy penalty associated to the formation of larger oligomers. Those of \mathbf{M}_{coop} are the typical characteristic distributions and free-energy profiles for cooperative supramolecular polymer systems, characterized by a nucleation-and-growth mechanism (which is not present in the non-cooperative \mathbf{M} system).

In Fig. S11c we also report the survival probabilities of different aggregate states (i.e. the probability that a monomer remains in the same state, along a time-interval of $\Delta\tau=15$ ns). This data highlights the qualitatively different behavior of the two models from the point of view of the dynamics/kinetics of the various size assemblies that populate the systems.

As already described above (see response to point 4 by the Reviewer), in this revised version of the SI, we also added further information to underline the cooperativity of \mathbf{M}_{coop} as opposed to the \mathbf{M} system. Apart from the already mentioned difference in dimerization free-energy (Fig. S12a-c, see response to point 4), showing that dimers are more stable in the \mathbf{M} system, we also verified how the free-energy of dimerization changes in \mathbf{M}_{coop} , when one of the dipole vectors is restrained to remain orthogonal to the monomer plane. This is intended to mimic the behavior of dipole vector orientation in a fiber tip monomer. As described above, these obtained free-energies (Fig. S12d, using WT-MetaD, see revised Methods), can be thus interpreted as a proxy for assessing the cooperativity in terms of non-constant (or super-additive) free-energy gain for monomer binding underpinning cooperative "elongation" in the \mathbf{M}_{coop} system. The free-energy increase with respect to dimerization is thus an additional demonstration of cooperativity.

In this revised version of the article, we now refer to these new additional results at page 7 line 21, where the \mathbf{M}_{coop} model is presented:

"The polymerization of longer oligomers/fibers, in which the dipoles at the ends are already aligned is thus favored compared to that of monomers/smallest assemblies, where dipoles have random or weak alignment and their interactions are inherently less stable and more dynamic (see Figure S12d). This confers to \mathbf{M}_{coop} the typical features of a cooperative supramolecular system, such as, e.g., the presence of a critical nucleus for fiber elongation, and a characteristic bimodal size distribution (see below)."

And later on at page 22 line 7, where the different response of the two \mathbf{M} and \mathbf{M}_{coop} systems is discussed:

"The difference in the ensemble behavior of cooperative vs. non-cooperative systems lies in the concept of super-additivity."

In a cooperative system, interactions are super-additive, promoting the growth of larger aggregates and unfavoring aggregates smaller than a critical size (see also the free-energy profiles in Figures S11-12, associated to the polymerization equilibrium of the two systems)."

Concerning the choice of the language, in the article we maintained the information on cooperativity and system thermodynamics in the form of probability distributions, because we wanted to highlight the monomeric population of the different aggregate states, and because this representation is also propaedeutic to the discussion of the transition matrices. Where the spotlight is on the inner transition dynamics of the supramolecular systems, we relied to quantities such as P_{poly} , P_{depoly} or Resilience, built on the transition probabilities of the monomeric states (defined in the Methods section). Finally, we would like also to underline how the use of "resource" as an additional term in this work, has the sole purpose of making the conclusions and implications of this work more clear and understandable for a broad readership: this is important, as we believe that some of the implications of these results indeed may be interesting also out the realms of supramolecular chemistry and (molecular) self-assembly.

Minor remarks

1. Plot color and label. In many figures, the post-stimuli systems are labeled by the stimuli concentration, i.e., 40C, 120C, ..., 400C, whereas the pre-stimuli (or no stimuli) systems are labeled as M and Mcoop for the two respective systems. The authors can make the label more consistent for better clarity.

Authors' response:

We have adjusted the figure legends to be more consistent.